# Agent-based modelling reveals feedback loops and non-linearity between mating system evolution and disease dynamics

Riccardo Tarantino[1*], Francisco Garcia-Gonzalez[2,3*]

1 Department of Humanities, University of Palermo, Palermo, Italy, 2 Doñana Biological Station (CSIC), Seville, Spain, 3 Centre for Evolutionary Biology, School of Biological Sciences, University of Western Australia, Crawley, Western Australia, Australia

* riccardo.tarantino01@unipa.it (RT); paco.garcia@ebd.csic.es (FG-G)

## Abstract

Despite its potential for providing a deeper understanding of both evolutionary processes and epidemic dynamics, the reciprocal interrelationship between mating system evolution and sexually transmitted diseases remains largely unexplored. Here we developed an agent-based model simulating the evolution of two different female mating strategies (monandry versus polyandry) under the spread of a hypothetical sexually transmitted disease implying reproductive costs. Our results strongly support the existence of feedback loops between the pathogen's transmissibility and the evolution of mating strategies. Importantly, we found several unexpected, non-linear emerging behaviours of the system, as well as tipping points which were undocumented under the conditions we considered here: i) medium-high/high probabilities of disease transmission per sexual contact reveal switches between disease-free and endemic outcomes, ii) counterintuitively, the disappearance of the pathogen might be a good indicator and predictor of the imminent extinction of the polyandrous genotype/phenotype/strategy from the population, and iii) probabilities of transmission above a medium-low threshold can offset the spread of polyandry even when this behaviour entails pronounced benefits. More broadly, our results illustrate that research into reciprocal influences between the dynamics of disease spread and sexual behaviour can provide valuable insights into disease transmission and the evolution of reproductive strategies, as well as into the sensitivity of mating systems' evolution to small variations in ecological and disease contexts.

## Introduction

Sexual behaviour, patterns of sexual interactions, and mating systems are extremely variable across sexually reproducing species [1–5]. In addition to genetic monogamy and social monogamy (where copulations outside the social pair-bond occur), the main types of mating systems include polygyny (males mating with different

**Data availability statement:** All relevant data are within the manuscript and its Supporting Information files. In particular, the NetLogo model can be opened by installing the NetLogo simulation platform (version 6.2.0), which can be downloaded free of charge from the official website of the software at the following link: https://ccl.northwestern.edu/netlogo/6.2.0/ or through the online viewer: https://www.netlogoweb.org/launch. Raw data generated by our NetLogo model have been uploaded to the Zenodo repository and are available at the following link: https://doi.org/10.5281/zenodo.17155761.

**Funding:** "This study was partially funded by a grant from Ministerio de Ciencia e Innovación (PID2019-105547GB-I00/MCIN/AEI/10.13039/501100011033), to FGG. URL of the funder's website: https://www.ciencia.gob.es/en/Ministerio/Mision-y-organizacion.html. The funder played no role in the study design, data collection and analysis, decision to publish, and preparation of the manuscript.".

**Competing interests:** The authors have declared that no competing interests exist.

partners), polyandry (females mating with multiple males), or polygynandry (polygamy where both males and females mate with multiple individuals of the opposite sex) [3,4]. Noticeably, each of these systems exhibits substantial variance surrounding mate acquisition and competitive reproductive success [3,4,6]. Such high levels of variability are patent not only across taxa but also, in some cases, within species. In humans, for example, anthropological studies indicate that all the mating systems mentioned above, and certainly many forms and diversity of sexual behaviours were, or are, present [7–10]. Heterogeneity in reproductive strategies within a same species has also been reported in non-human animals. Aside the existence of alternative male mating strategies in a variety of animals [11,12], in some butterflies and beetles monogamous and polyandrous female phenotypes can be found in a same population [13], and in some species (e.g., of fish and mammals) interpopulation variability in social and genetic mating systems has been documented [14,15].

There is an increasing realization that heterogeneity around the patterns of variation in reproductive success and sexual behaviour is explained by environmental conditions and that the evolution of mating systems and male-female evolutionary dynamics (including sexual conflict and sexually antagonistic selection) is dependent on ecological and demographical factors. For instance, temperature [14,16,17], diet or dietary stress [18,19], or other ecological and social factors [reviewed in 6] alter the opportunity for sexual selection, which on top of exhibiting such context dependencies is also temporally dynamic [20]. Furthermore, factors such as population spatial structure, the thermal environment or the broad ecological context are known to affect sexual conflict and sexually antagonistic coevolution [16,21–25].

Another factor that has the potential to shape sexual interactions and sexual selection is the presence of sexually transmitted diseases (STDs). However, despite it is presumed that pathogenic infections may affect mating dynamics, the effects of STDs on mating system evolution are still poorly known [26–30]. Similarly, whether mating system and sexual behaviour variation determine epidemiological dynamics has long been hypothesised, since it seems logical to expect that the higher the number of sexual interactions between individuals the greater the risk of STD transmission [27,28]. However, final outcomes are complex and may depend on a variety of factors relating to pathogen transmission and virulence, as well as on the benefits and costs of the different sexual and reproductive strategies [27,29,31]. Despite this complexity, both sexual behaviour and STDs evolution are likely governed by mutual effects. Investigating the coevolution or feedback loops between epidemiological and mating system dynamics is bound to be key to understand disease transmission and sexual behaviour in sexually reproducing species, including humans, but whether these feedback loops indeed exist, and in the case they exist, the conditions that may promote them, remain largely unexplored [29,32,33].

Existing work on the concurrent effects of sexual behaviour and the dynamics of sexually transmitted infections highlights the importance of specific conditions underlying such coevolution and illustrates that important gaps need to be addressed to fully understand the mutual interplay. Ashby and Gupta [32] investigated the impact of polygamous mating systems on disease incidence and pathogen virulence and

found that less virulent strains may be favoured in highly skewed mating systems. They also found a positive relation between the degree of polygamy and the number of serial monogamists affected by the infection. These results help to understand the connection between mating system and STDs in groups such as birds and humans, where serial monogamy is common. Other approaches have been illuminating as well. For instance, McLeod and Day [27] implemented a model with an STD as the only selective force, simulating the invasion of a monogamous mutant in a promiscuous population and determining the selective advantages of monogamy in relation to several transmission rates. On the other hand, Bauch and McElreath [34] addressed the emergence of socially imposed monogamy in humans through simulations of the diffusion of bacterial STDs. Much of the previous research focusing on the interplay between mating systems and STDs involves the implementation of network models. For instance, in Ashby and Gupta's [32] study, sexual contact networks were used to evaluate the effects of STDs in a polygamous context. Several studies in humans focus on nodes connectivity resulting from social structures and preferences underlying sexual intercourse [35–38]. Some networks also consider further details, like the influence of temporal structures [39]. Alternatively, agent-based models not relying on networks can also involve both temporal and spatial components, so they are also suitable tools to simulate the evolution of systems of biological and epidemiological interest [40]. These models are based on computational individuals (the agents) with a well-defined set of characteristics and an algorithmic behaviour, and whose multiple interactions determine the global dynamics of the simulated system.

In this article, we use an agent-based model to simulate the effects of a STD on the evolution of mating strategies, which was assessed through cross-generational changes in the frequency of reproductive strategies and underlying changes in the frequencies of alleles that determine such sexual behaviours. At the same time, we simulate the effects of mating strategy on epidemic dynamics. In doing so, we also introduce some aspects of the interplay between STDs and mating systems that have been overlooked in previous assessments. For instance, we implement simultaneously monandry and several degrees of polyandry, and this while keeping general benefits and costs of mating separated from infection-related costs. We also consider that females are not always receptive to mating partners, e.g., because of non-receptivity or because they are giving birth or ovipositing (i.e., we implement a time out of the mating pool [41,42]). Another original angle of our approach is that it explicitly considers some typical characteristics of complex systems such as tipping points and non-linearity underlying bridges between evolutionary dynamics and epidemiology. Based on our computational results, we contend that a profound, reciprocal, and sometimes not linear, evolutionary interrelationship between sexual behaviour and the dynamics of sexual pathogen infections may be a common phenomenon in natural systems. These results, therefore, have important implications for understanding both mating system evolution and the variation in sexual strategies, as well as the epidemics of STDs.

## Methods

### Model overview

We used NetLogo version 6.2.0 [43] to implement our agent-based model (S1 Appendix), and its integrated tool *Behavior-Space* to run a set of different scenarios. For more details on the settings we considered in our study, see the subsection *Experimental sets* below. The model simulates the evolution of two alternative mating strategies adopted by females of a hypothetical animal species, i.e., monogamous ("M") and polyandrous ("P") (these letters are also used to refer to each type of female). The key features of the model are applicable to a wide range of taxa with sexual reproduction, including many invertebrate and vertebrate species (see below). Simulations can be run either without any circulating pathogen or considering an initial number of infected individuals which may infect susceptible individuals of the opposite sex during copulation.

As for the two alternative mating strategies, we modelled them as genetically inherited behavioural traits, which is an element of biological realism, as shown by several case studies [13,44–47]. More specifically, in our model, each female

strategy is considered as a behavioural phenotype corresponding to one of two possible genotypes. Each female offspring inherits her genotype from her mother, so all females generated by M mothers are also M, while all females generated by P mothers are P. All males are polygynous, regardless of maternal genotype. Males' polygyny may be considered a very widespread condition in nature, as shown by the massive literature on this subject [1,3]. Given the genetic inheritance of these behavioural phenotypes and the complete penetrance of the hypothetical alleles determining them (i.e., phenotype is not affected by epistasis or environmental variables), all agents behave consistently through their entire life, in accordance with the genotype they inherit. During simulations, agents randomly move on the *NetLogo* lattice, update their age, mate, produce offspring in different batches, and die due to age or random extrinsic mortality. Mating between two adult, sexually receptive individuals of opposite sex is random and is modelled as an event of co-localization on the same coordinates.

As for the sexually transmitted pathogen, it can be considered either a virus, a bacterium, a fungus, or a protozoan interchangeably, since we are only interested in its effects on fitness of the simulated animals. For the same reason, we did not create a further class of agents for the pathogen. In fact, we implemented the infection as a state: agents which have contracted the pathogen are infected, whereas agents that have never been infected and agents that have recovered from the infection are susceptible. Indeed, simulations can be run either in SI (i.e., Susceptible-Infected) or SIS (i.e., Susceptible-Infected-Susceptible) mode. In a SI scenario, agents which have contracted the pathogen remain infected and contagious for their entire lifetime, whereas in a SIS scenario agents recover from the infection after a given time interval, then they become susceptible again. In other words, recovered individuals do not gain any immunity against the pathogen. The infection can be transmitted from an infected agent to a susceptible one via mating, with a given probability of transmission which is applied at each sexual contact. This is the case of horizontal transmission of the pathogen. Although most of our experimental scenarios do not include vertical transmission of the infection, that is, from parents to their offspring, in some specific scenarios we implemented vertical transmission, with infected mothers transmitting the disease to their newborn offspring generated while they are infected. Finally, infection implies a reproductive cost for the host: everything else being equal, infected individuals generate less offspring than susceptible individuals. It is worth noting that our focus is on the costs of infection in terms of reproductive success. In evolutionary terms, what matters most is how infection affects reproductive success. Our model reflects this by linking infection to reduced offspring production, which directly represents a loss in fitness. This summarising perspective allows broader generalisation.

All stochastic processes and events involved in our computational system, including movement, mating, death, and probability to contract the infection after copulation, were modelled using the NetLogo primitive *random* and a few other closely related terms, which rely on the Mersenne Twister algorithm and thus generate uniform pseudo-random numbers with a very long period [48]. These functions are essential to preserve the probabilistic nature of complex ecological and evolutionary systems and ensure that each run gives a different outcome based on the same input values.

While our model is designed to provide insights across various case studies and taxa, it inherently possesses limitations common to models of complex biological systems. We point out the potential applications and main restrictions of the model in subsection *Model applicability and limitations* below.

## Parameters

The parameters included in the agent-based model were designed to achieve a reasonable compromise between generality and biological realism, so we can consider the tendencies shown by our outcomes as applicable to a good variety of real-world cases. Information on the parameters that we implemented is synthesised in Table 1.

Basic variables regard the initial population size, its composition in terms of males, polyandrous females and monogamous females, longevity, and probability of death due to causes other than age. The total number of agents at the beginning of a simulation varies from a few to several hundred, and it is also possible to consider unbalanced ratios between sexes and/or alternative mating strategies. Longevity depends on the parameter *Lifespan*, ranging from 0 to 1000 ticks,

**Table 1. Parameters included in the NetLogo model, including a short description, the range of all possible values selectable in the interface of the model, and references.**

| Parameter | Description | Range | References |
|---|---|---|---|
| Initial-Males | Number of males at the beginning of the simulation | 0 - 200 | – |
| Initial-Mfemales | Number of monogamous females at the beginning of the simulation | 0 - 100 | – |
| Initial-Pfemales | Number of polyandrous females at the beginning of the simulation | 0 - 100 | – |
| Lifespan | Age of death of agents (in ticks; see text) | 0 - 1000 | – |
| Juvenile-Threshold | Maximum number of young, non-reproductive agents for each tick (i.e., a random mortality for exceeding zygotes/eggs/newborn individuals is introduced) | 0 - 2000 | [49–51] |
| Age-of-Sexual-Maturity | Age at which agents begin to be available for mating (in ticks; see text) | 0 - 100 | – |
| Max-Matings | Maximum number of matings with different males a polyandrous female can have | 2 - 5 | [44,52–54] |
| Time-Out-of-the-Market | Sexually non-receptive period (in ticks; see text) assigned to each polyandrous female after mating. For monogamous females, it is the time between each oviposition/birth event and the next one. | 0 - 50 | [55–57] |
| Benefit-of-Mating-%* | Increase in the total number of offspring generated by each female. It is applied one time for each mating (e.g., a 4% value causes a polyandrous female who mates 2 times to produce 4×2=8 more offspring) | 0% − 10% | [51,58–60] |
| Cost-of-Mating-%* | Decrease in the total number of offspring generated by each female. It is applied one time for each mating (e.g., a 2% value causes a polyandrous female who mates 5 times to produce 2×5=10 less offspring) | 0% − 10% | [61–65] |
| Index-Cases | Number of agents which are already infected at the beginning of the simulation | $0 − N$, where $N$ is the total number of agents in the simulation | – |
| Probability-of-Transmission-% | Probability of contracting the disease applied to each agent at each sexual contact (i.e., each time an adult male and a receptive female share the same coordinates in the NetLogo lattice) | 0% − 100% | – |
| Offspring-Reduction-%* | Reproductive costs of infection. Decrease in the total number of offspring generated by each female affected by the pathogen and/or whose partner is affected by the pathogen (e.g., a 1% value causes a mother to produce 1 less offspring if she is infected, and 1 less offspring for each mating with an infected male) | 0% − 10% | [26,66–68] |
| Time-to-Recover | Time required for an agent to become susceptible again after being infected (only in the SIS version of the model and expressed in ticks; see text) | 0 - 100 | [69–71] |
| Model-Version | The user can select either the SI or the SIS version of the model | – | – |
| Infected-Offspring? | The offspring of an infected female can be either all susceptible or all infected depending on this switch (i.e., it enables/disables vertical transmission of the infection) | 0/1 | [26,72] |

*All variations in the total number of offspring are applied to a basal number of 100. In other words, 100 is the fixed number of offspring produced by each female in absence of benefits or costs in terms of fecundity.

i.e., the unit of measure of time in NetLogo simulations. There are several alternative ways to interpret ticks in our model: for instance, when thinking about short life cycle species (e.g., insects and other invertebrates), one could consider each 10 ticks to be equivalent to 1 day, while other taxa (e.g., reptiles or mammals) would be also represented by considering each block of 10 ticks as the equivalent of 1 month, or 1 year, or any appropriate length of time. Agents start with age 0 and are non-sexually mature until they reach the age indicated by the parameter *Age-of-Sexual-Maturity*. Before that age, they can only randomly move around the simulation world. After that point, they start mating with random partners in a panmictic population. Finally, when their age is equal to the *Lifespan* value, they die. In all our simulations, we used a *Lifespan* of 150 ticks (i.e., 15 days/months/years) and an *Age-of-Sexual-Maturity* of 10 ticks (i.e., 1 day/month/year) to keep a realistic ratio between adulthood and juvenile period that could be applied to many species. For instance, in the case of mammals, the median longevity estimates of some dog breeds are around 13 years, while puberty is reached

between 6 months and 2 years of age [73,74]. As for the parameter *Juvenile-Threshold*, it is used to limit the maximum number of co-existing young, non-reproductive individuals, so it applies random extrinsic mortality to zygotes/eggs/newborns. Indeed, we assume that mortality is much greater for zygotes and juveniles than for adults [49–51].

Some other parameters are strictly inherent to mating and reproduction. As a basic assumption, we consider a fixed numerical value (100 offspring) to indicate the typical baseline lifetime fecundity of each female in the hypothetical species. This number can be affected by a series of factors. *Benefit-of-Mating-%* is a percentual increase of the number of offspring based on the number of matings of each female. This parameter was introduced into the system to incorporate direct benefits of mating [58,75], that translate into fecundity increases. Conversely, *Cost-of-Mating-%* is a percentual decrease that is also based on the number of matings and that allows to consider situations where mating is costly [61,62,64,65,76]. The total number of offspring ($N_o$) for each female is given by equation (1):

$$N_o = 100 + N_m \Delta m, \tag{1}$$

where $N_m$ is the number of matings realized by the female, and $\Delta m$ is the difference in percentage between benefits ($b$) and costs ($c$) of mating. For instance, if a M female mates in a context where $b$ is set at 4% and $c$ is set at 3%, her number of offspring is: 100 + 1(4–3) = 101 offspring. Similarly, the number of offspring generated by a P female mating 4 times would be: 100 + 4(4–3) = 104. Therefore, parameters related to benefit and cost of mating determine the total number of offspring generated by each female during her whole lifespan. The total number of offspring generated by each female, regardless of her genotype, is always reached at the end of several oviposition/birth events separated in time. In particular, M females generate 3 separate batches of offspring, while P females generate a variable number of separate batches, which is equal to their individual degree of polyandry. The degree of polyandry (i.e., a female's number of mates) is another relevant, well-acknowledged variable associated with polyandry [44,52–54,77], and it is implemented in the model with the parameter *Max-Matings*. This parameter indicates the maximum possible number of matings for P females, ranging from 2 up to 5. For instance, if it is set to 2, this means that all P females mate twice, while if it is set to 4, this means that P females mate from 2 to 4 times (determined randomly for each individual P female). Consequently, we introduced in our model variation around the degree of polyandry. As for the time required between an oviposition/birth event and the next, it is expressed in ticks and directly influenced by the parameter *Time-Out-of-the-Market*. During this interval of time, which starts immediately after mating, females are not available for further sexual intercourses. In the model, it is applied in a slightly different way to M and P females. For M females, it is the time required between each oviposition/birth event and the next, while for P females is the time required to restore sexual receptivity after mating. This non-receptive period is a known phenomenon in the real world [55–57], and, for instance, it may be associated to the time each female dedicates to maternal care. The *Time-Out-of-the-Market* parameter covers values from 1 up to 50 ticks. Although this value is regulable, we chose to keep it fixed at 6 ticks for all our simulations because we found that this interval is appropriate to ensure longer coexistence and equal probabilities of fixation for each mating strategy, everything else being equal. For further details on this baseline condition, see the Results and Discussion section.

All remaining variables are crucial to introduce the epidemic dynamics into the system. The parameter *Index-Cases*, ranging from 0 up to the population size, indicates the number of agents which are already infected at the beginning of the simulation. A limited initial number of index cases (e.g., 10, as in our experiments) may simulate a situation where a small group of individuals migrated from outside into a susceptible population. As for the transmission of the sexual disease and its impact on fitness, we implemented the parameters *Probability-of-Transmission-%* and *Offspring-Reduction-%*, respectively. *Probability-of-Transmission-%* is the probability that a susceptible agent contracts the infection immediately after mating with an infected agent of the opposite sex (i.e., when they are both available for mating and share the same coordinates). *Offspring-Reduction-%* is a cost which is independent from *Cost-of-Mating-%*, as it is the direct consequence of the disease on fertility or fecundity [26,66–68]. More specifically, this parameter indicates a percentual decrease in the

total number of offspring generated by females. In an epidemic context, a female at her last mating/oviposition/birth event would have generated a $N_o i$ number of offspring, as reported by equation (2):

$$N_o i = 100 + N_m \Delta m - (N_i + i)\,o, \qquad (2)$$

where $N_i$ is the number of matings with an infected male, $i$ is the binary value of the status of the female (1 if infected, 0 if susceptible), and $o$ is the percentual reduction of offspring induced by the infection. It is important to notice that even a susceptible female may incur a reduction in offspring due to the disease condition affecting one or more of her partners, even if she does not contract the infection (which depends on the probability of transmission). For example, let us consider a context where $b$ is 2%, $c$ is 3%, and $o$ is 1%. If an infected M female mates with a susceptible male, she would produce a total of $100 + 1(2-3) - (0+1) \times 1 = 98$ offspring. If a susceptible P female mates three times (two of the times with infected partners, and one time with a susceptible one) but does not contract the infection during her lifetime, e.g., due to low values of *Probability-of-Transmission-%*, she generates a total of $100 + 3(2-3) - (2+0) \times 1 = 95$ offspring. In a SIS context, the status of an infected female ($i$) is set back from 1 to 0 after *Time-to-Recover*, and the total number of offspring generated by this female is calculated according to the status change. Finally, if a switch enabling the possibility of vertical transmission of the infection from infected mothers to their offspring is turned on (*Infected-Offspring?*), these females will only produce infected offspring as long as they are also infected.

An additional feature of our model is represented by a counter that keeps track of all sexual contacts of polyandrous females during simulations. The counter appears in the NetLogo interface as a monitor called *Sexual Contacts (P Females)* and has been useful to explore some aspects of our simulations in greater depth (see the Results and Discussion section).

## Model applicability and limitations

We use SI and SIS epidemic models of a vertically or horizontally sexually transmitted disease that can be applied to various diseases and infections in groups as diverse as humans, mammals, reptiles, birds and arthropods (see Fig 1).

Our baseline conditions (see below) relate to an infection-free system with an equal sex ratio, a maximal number of 5 matings per female, high fecundity (100 offspring), sexual maturity reached once the first 7% of lifespan has been consumed (10 ticks -the units of measure of time in the simulations- for a lifespan of 150 ticks), and time to restore receptivity after mating equivalent to a 4% of lifespan time (time out of the market = 6 ticks). These values were chosen for two reasons. First, they can be applied to a variety of animals and life-cycles, from short-lived to long-lived species (depending on the presumed duration of each time unit in our system) [85–87], with variable but moderate female remating/multiple mating rates that are common across animal mating systems [44,52,88–95]. Second, they result in balanced outcomes supporting the coexistence of both mating strategies over time, provided there is no variance in the costs and benefits of mating, infection costs, or transmission probability—factors whose variations and effects on mating systems and infection dynamics are the focus of our study.

Models on the interplay between mating system and STDs frequently consider fertility or fecundity costs of disease [29,30,96]. Indeed, STDs typically cause such reproductive reductions in hosts. For example, Lockhart et al. [78] and Smith and Dobson [79] discuss several cases of STDs with severe effects on fertility in a variety of taxa including mammals, snails, and birds, while Webberley and Knell [26] list no less than 14 STDs across insects that lead to pathologies including reduced fertility, damaged sperm, decreased hatch rates, gonadal hypertrophy or decreased fecundity. Furthermore, physiological costs associated with immune defence are expected, and infections may reduce reproductive output through trade-offs between resource allocation to immunity defence and reproductive effort [97].

Our model, like all models, simplifies the real world, and thus, our results need to be considered as approximations. In real-world scenarios, a multitude of environmental, social, ecological, and demographic factors can significantly influence

**Fig 1. Some examples of sexually transmitted diseases and infections in animals and humans, with indication of their mode of transmission (vertical or horizontal) and applicability to SI (susceptible-infectious) or SIS (susceptible-infectious-susceptible) epidemiological models.** Information collected from various sources including [26,78–84]. The examples include vector-transmitted diseases that are sexually transmitted in the vectors themselves (e.g., La Crosse virus, dengue fever and other flavivirus, spotted fever, tick-borne encephalitis, Crimean-Congo haemorrhagic fever or the African swine fever); some of these infections are not only costly for the hosts but also the vectors. [a]STD predominantly horizontally transmitted, but mother-to-child infection can occasionally occur during pregnancy or birth.

the costs and benefits of mating and STD dynamics. For instance, sex ratio allocation and biases, social structure, social status and social dominance, population density, spatial/geographical features, resource quality and availability and temperature, amongst other factors, are known to modulate sexual interactions [6,16,23,98–102], and, consequently, they would be expected to influence disease epidemics to some extent. However, we do not include all the variables possibly influencing the patterns. Our goal is to focus on specific variables influencing mating system evolution and disease dynamics, allowing us to explore general trends and non-linear patterns that may be shared across different systems. This approach enhances the model's tractability and usefulness, allowing for clearer insights into the primary factors influencing the patterns studied [103].

Our approach has additional, more specific, limitations. For instance, while in our SIS model, infected individuals can recover from infection after a regulable time interval, they do not have immunity after recovery (i.e., recovered individuals can be reinfected immediately after recovery). Nevertheless, this is the case for some STDs, including some sexually transmitted infections in humans.

### Experimental sets

We used the model to run several sets of simulations with a variety of different experimental settings. We conducted Symmetric Analyses simulations, henceforth "SA", where the initial number of M and P females is the same, that is, 50 agents

for each strategy (see the different sets of SA simulations in Table 2). We also conducted Invasibility Analyses simulations, henceforth "IA", where 1 M female invades a population of 99 P females (see Table 3). Furthermore, we conducted simulations with additional variations that were designed to test some aspects of the interplay between female mating strategy and the STD when populations lacked polymorphism regarding female mating genotype (see below and Table 4).

To decipher the weight of the interplay between different mating strategies and epidemic dynamics, we focused mainly on the impact of our parameters on the gene pool of the population. Since almost all simulations end with the irreversible fixation or extinction of one genotype/mating strategy (i.e., an absorbing state), we considered the frequency of fixation and the mean time to fixation as the key values for monitoring the evolution of the system. Both frequency and time (in ticks) can be extracted from the terminal time step of each simulation, so in these cases we did not examine the behaviour of the system tick after tick using *BehaviorSpace*. The exceptions are the sections/experiments "*Who carries the pathogen*" and "*Co-occurrences of extinction of the pathogen and loss of the P phenotype*", which were based on samples of 30 and 250 simulations, respectively, and whose numerical values were reported tick after tick, not just at the end of each replication (see below).

In simulations where the disease spread was considered, *Index-Cases* were set at 10 to ensure that at least some of the initial infected agents could eventually survive the phenomenon of random extrinsic mortality affecting juvenile individuals and reach the adult age. When the pathogen was present in the system, we always simulated situations where *Offspring-Reduction-%* is 1%, to avoid unrealistic high costs for P females. When using the SIS version of the model, we set *Time-to-Recover* = 50 ticks (i.e., 5 days/months/years), which is a moderate duration of the disease compared to the lifespan of agents in our scenarios.

Sex ratio was always kept constant at 1:1 from the beginning of each simulation. We decided not to run simulations with deviations from equal sex ratio because preliminary assessments suggested that tilting the balance had no relevant, long-lasting effects on the evolution of the digital population. Finally, the values of *Age-of-Sexual-Maturity, Lifespan*, and *Time-Out-of-the-Market* were kept constant throughout all simulations (see Tables 2, 3, and 4), the reason being that, everything else being equal, these values lead to balanced and overall stable "neutral" outcomes (i.e., our baseline condition). We confirmed in preliminary runs that such settings lead to the coexistence of both mating strategies along time in the absence of variance in the cost and the benefits of mating, the costs of infection, and the probability of transmission, which are the factors whose variation effects upon mating system and infection dynamics are targeted in our study. For analogous reasons, we also used a single value for *Juvenile-Threshold*, although the dynamics generated by the model seem to undergo slight variations when very low values are assigned to this variable (see Results and Discussion).

In the experimental scenario "*Who carries the pathogen*", we extracted the proportions of infected P and M females from the 10th tick (i.e., when the individuals of the first generation become fertile) up to the last tick with at least one infected female of each mating strategy. This restriction was necessary, as beyond that point, the presence of only one phenotype would make it the sole carrier of the infection by default, preventing a meaningful comparison of which strategy was most affected by the disease. Similarly, as for the experimental scenario "*Co-occurrences of extinction of the pathogen and loss of the P phenotype*", the correlation coefficient between the time needed for P females' extinction and the time needed for the end of the epidemics was calculated only considering simulations where both P females and infected agents ultimately disappeared. This way, we excluded outcomes where the P strategy continued spreading for a long time (i.e., at least 5000 ticks from the beginning of the simulation) in absence of the STD. In other words, we were interested in calculating the correlation as long as both the P genotype and the STD were present in our system because our aim in this scenario was to focus *only* on simulations where polyandry and the epidemics share the same evolutionary fate.

## Results and Discussion

### System's response to variations in costs of mating

Simulations of the first scenario were carried out on a population whose individuals experienced fixed *Benefit-of-Mating-%* (3%) and *Cost-of-Mating-%* varying from a minimum of 1% up to a maximum of 5% for each mating (Fig 2A and Fig 2B).

**Table 2. Settings in experiments using Symmetric Analysis (SA).**

| Experimental Scenario (SA)* | Juvenile-Threshold | Lifespan | Age-of-Sexual-Maturity | Cost-of-Mating-% | Benefit-of-Mating-% | Maximum-Matings | Time-Out-of-the-Market | Index-Cases | Probability-of-Transmission-% | Offspring-Reduction-% | Time-to-Recover | Model-Version | Infected-Offspring? |
|---|---|---|---|---|---|---|---|---|---|---|---|---|---|
| Baseline condition | 1500 | 150 | 10 | 3 | 3 | 5 | 6 | 0 | 0 | 0 | N/A | SI | 0 |
| Variable costs | 1500 | 150 | 10 | 1/2/3/4/5 | 3 | 5 | 6 | 0 | 0 | 0 | N/A | SI | 0 |
| Who carries the pathogen | 1500 | 150 | 10 | 3 | 3 | 5 | 6 | 10 | 100 | 0 | N/A | SI | 0 |
| Variable transmissibility | 1500 | 150 | 10 | 3 | 3 | 5 | 6 | 10 | 70/85/100 | 1 | N/A | SI | 0/1 |
| Co-occurrences of extinction of the pathogen and loss of the P phenotype | 1500 | 150 | 10 | 3 | 3 | 5 | 6 | 10 | 100 | 1 | N/A | SI | 0 |
| High benefits, variable transmissibility | 1500 | 150 | 10 | 4 | 5 | 5 | 6 | 10 | 25/50/75/100 | 1 | 50 | SIS | 1 |

*In all these scenarios, *Initial-Males* = 100, *Initial-Mfemales* = 50, and *Initial-Pfemales* = 50. The number of runs performed for each setting of these scenarios is 500, except for the scenario "Who carries the pathogen" (30 runs) and the scenario "Co-occurrences of extinction of the pathogen and loss of the P phenotype" (250 runs).

**Table 3. Settings in experiments using Invasibility Analysis (IA).**

| Experimental Scenario (IA)* | Juvenile-Threshold | Lifespan | Age-of-Sexual-Maturity | Cost-of-Mating-% | Benefit-of-Mating-% | Maximum-Matings | Time-Out-of-the-Market | Index-Cases | Probability-of-Transmission-% | Offspring-Reduction-% | Time-to-Recover | Model-Version | Infected-Offspring? |
|---|---|---|---|---|---|---|---|---|---|---|---|---|---|
| Baseline condition | 1500 | 150 | 10 | 3 | 3 | 5 | 6 | 0 | 0 | 0 | N/A | SI | 0 |
| Variable costs | 1500 | 150 | 10 | 1/2/3/4/5 | 3 | 5 | 6 | 0 | 0 | 0 | N/A | SI | 0 |
| Variable transmissibility, large cost-benefit gap | 1500 | 150 | 10 | 5 | 2 | 5 | 6 | 10 | 25/50/75/100 | 1 | 50 | SIS | 1 |

*In all these scenarios, *Initial-Males*=100, *Initial-Mfemales*=1, *Initial-Mfemales*=1, and *Initial-Pfemales*=99. The number of runs performed for each setting of these scenarios is 500.

**Table 4. Settings in additional experiments considering populations containing P females or M females only.**

| Experimental Scenario* | Juvenile-Threshold | Lifespan | Age-of-Sexual-Maturity | Cost-of-Mating-% | Benefit-of-Mating-% | Maximum-Matings | Time-Out-of-the-Market | Index-Cases | Probability-of-Transmission-% | Offspring-Reduction-% | Time-to-Recover | Model-Version | Infected-Offspring |
|---|---|---|---|---|---|---|---|---|---|---|---|---|---|
| Minimum probability of transmission for endemization | 1500 | 150 | 10 | 3 | 3 | 5 | 6 | 10 | 25/50/60/70/80/90 | 1 | N/A/50 | SI/SIS | 0 |
| Persistence of the epidemics in a M population | 1500 | 150 | 10 | 3 | 3 | 5 | 6 | 10 | 100 | 1 | N/A | SI | 0 |

*In the first scenario, *Initial-Males*=100, *Initial-Mfemales*=0, and *Initial-Pfemales*=100. In the second scenario, *Initial-Males*=100, *Initial-Mfemales*=100, and *Initial-Pfemales*=0. The number of runs performed for each setting of these scenarios is 100.

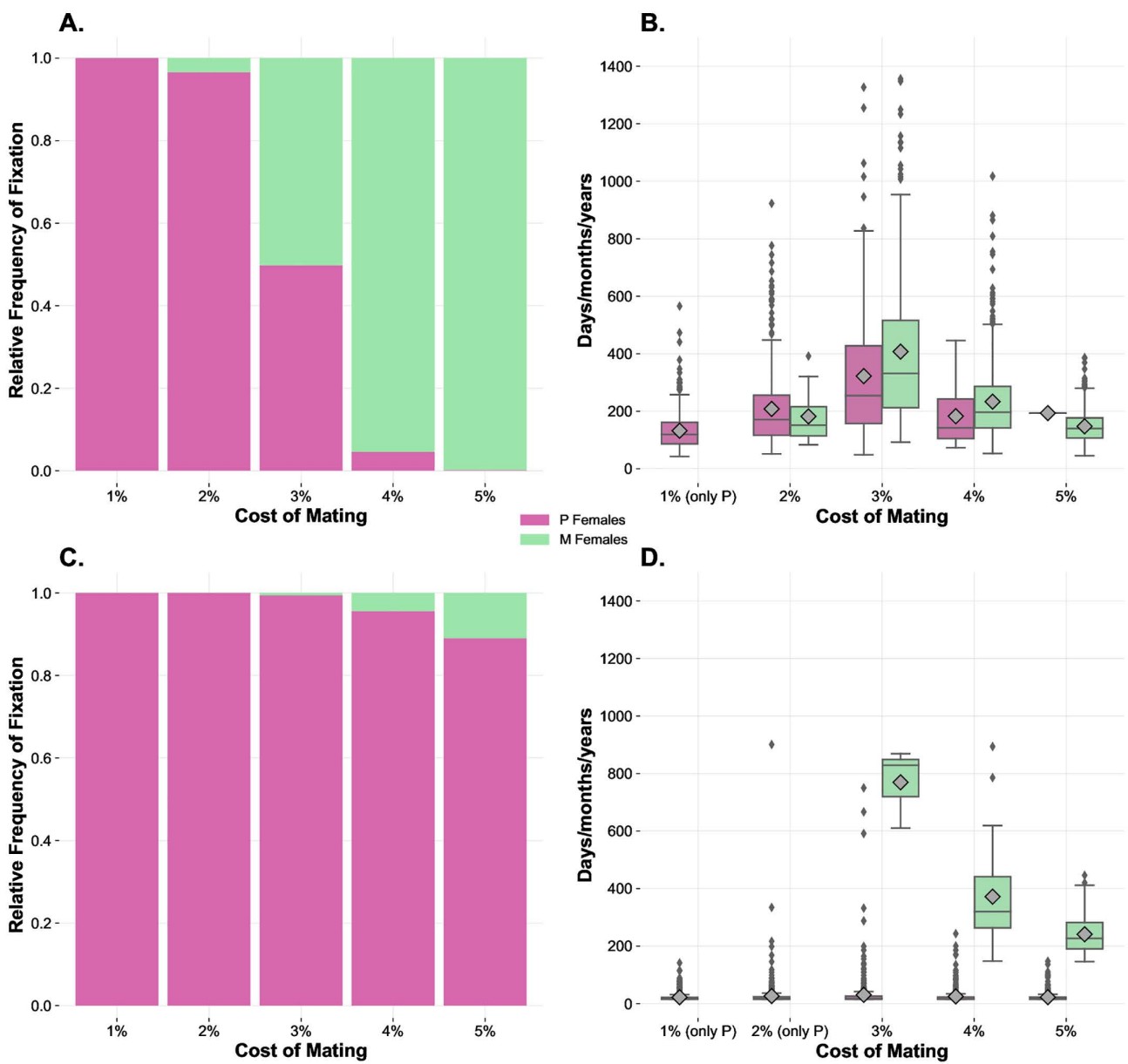

**Fig 2. The effects of variation in costs of mating on the relative frequency of fixation and time to fixation.** Frequency and times to fixation in SA (panels **A** and **B**, respectively) and IA analyses (panels **C** and **D**). The SA involves 200 initial individuals, 1:1 sex ratio, even distribution of female genotypes/mating strategies, fixed *Benefit-of-Mating-%* at 3%, and variable *Cost-of-Mating-%* (1%−5%). All individuals are susceptible and reach the adult age at 10 ticks. The IA has the same settings as the SA above but simulates situations where 1 monogamous female invades a population consisting of 99 polyandrous females. The case with equal benefit and cost of mating (3%) is the baseline condition. Relative frequency of fixation values (**A**, **C**) indicate the number of simulations where each genotype reached fixation out of the total number of simulations (500). Mean time to fixation values (**B**, **D**) indicate the mean time each genotype took to reach fixation and are represented by the grey symbols inside the boxplots. Paired boxplots provide information on the distributions of the times to fixation obtained.

Results under these conditions indicate that, as expected by our design (and as intended, to set up a system with neutral baseline conditions), benefits and costs of mating that are of similar magnitude cancel each other, and the larger the cost of mating (i.e., the cost of polyandry) the less likely P goes to fixation. For instance, a cost-benefit difference in

two percentual points (i.e., a fecundity cost of around 2%) was enough to lead to extinction of the P strategy in less than 2000 ticks (i.e., 200 days/months/years -depending on the unit of time attributed to each 10 ticks-, for animals having a life expectancy of 15 days/months/years, respectively). In other words, this outcome indicates that in our system the P strategy would become extinct in less than 200 years for a species in which animals live around 15 years, if the costs of polyandry exceed the benefits of multiple mating in just 2 percentual points. These results inform that even few percentual points of difference in the cost-benefit balance could produce remarkable effects on the system, ultimately eliminating the genetic polymorphism within the population in a relatively short interval of time.

Fig 2C and Fig 2D show the relative frequencies of fixation and times to fixation in the context of an invasion by an M strategist, that is, for instance, when in a population of P strategists a M mutation arises, or when an immigrant M individual arrives to a P population. These results indicate that costs of mating that were not compensated by the benefits of mating led to the complete replacement of the P genotype by the M genotype in approximately 5–10% of all simulations performed under these conditions (i.e., 5–10% fixation cases out of a total of 500 runs for costs of mating of 4% and 5%, when benefits of mating were fixed at 3%). The outcomes inform that even costs of mating which are slightly higher than benefits are enough to tilt the balance in favour of monogamy, even in the context of a considerable initial numerical minority for M strategists.

The results of this first scenario allow us to make some important preliminary considerations. Our first objective in this study was to create a computational model useful to infer meaningful patterns relating to the interrelationships between mating system evolution and epidemic dynamics. To this end, we first needed to find baseline conditions reflecting an equilibrium between the coexistence of M and P strategies. Such conditions were confirmed by a series of results, including those presented in this subsection and relating to the effects of variation in the cost-benefit balance of mating interactions. Focusing on the results of the SA with variable *Cost-of-Mating-%* (Fig 2A and Fig 2B), a positive relation between frequency of fixation of the M phenotype and costs of mating was identified. Conversely, a negative relation between frequency of fixation of the P phenotype and the costs of mating was observed, as expected. When benefits and costs of mating were equal (3%), the frequencies of fixation of the two strategies were similar, and the times of coexistence before collapsing to an absorbing state were longer. For these reasons, the latter settings were used to generate a baseline condition used as an equilibrium point whose disruptions were studied following the introduction of variance in the targeted parameters. As for the related IA (Fig 2C and Fig 2D), no fixation events for the M phenotype were observed below the threshold of 3% *Cost-of-Mating-%*. With costs at 4% and 5%, the M phenotype reached fixation approximately in 5–10% of the observed cases. At the same time, cost increases seemed to reduce the time to fixation of the M strategy, without affecting the time to fixation of the P phenotype. In sum, SA and IA informed that P females gained a clear advantage with lower costs of mating, while the M genotype proliferated when multiple mating was more penalised. These sets of results confirmed that we were successful in generating a system recreating the importance of variations in costs and benefits of mating [59,104–106]. Indeed, the primary aim of this basic scenario was to test whether our model can generate largely expected outcomes which are in line with experimental findings related to the effects of variation in costs of mating. More specifically, it is well-known that a high cost of mating severely acts against the spread of a P strategy [104,107–109], which is a pattern confirmed by our results. Therefore, this scenario is especially important as it allowed us to validate our agent-based model, which can be considered reliable concerning its ability to simulate the consequences of costs and benefits of mating on P and M strategies.

### Epidemic dynamics

In the experiment that we termed "*Who carries the pathogen*" we inspected the interplay between epidemic dynamics and mating strategies by looking at whether the spread of the pathogen was linked to the evolutionary trajectories of a particular strategy, and with what intensity. To do this, we monitored the proportion of infected P females and infected M females over the course of 30 simulations, starting from tick 10 (i.e., when agents reach their sexual maturity) until the last

tick in which at least one infected female was present for each mating strategy. The expectation that P females would be affected more than M females by the STD due to their higher numbers of sexual contacts was strongly supported. Indeed, the distributions of proportions of infected P females were predominantly centred at around 80% in the majority of the simulations performed, as shown in Fig 3.

Fig 4 shows a representative example from a single run, which makes clear that the infection mainly affects P females, in this case for the entire duration of the run. The fact that P females are the main carriers of the infection is also suggested by outcomes from an additional experiment ("*Persistence of the epidemics in a M population*"), which was carried out using a homogeneous M population: in this scenario the pathogen, which never became endemic in any of the 100 simulations, always disappeared soon after the beginning of each run.

To sum up, we found that P females are much more affected by the STD than M females, so the former are the main vehicles for the transmission of the disease, a result echoing previous findings [78,110]. Based on this observation, numerical increases of the P phenotype would be related to a greater diffusion of the infection in the population, since

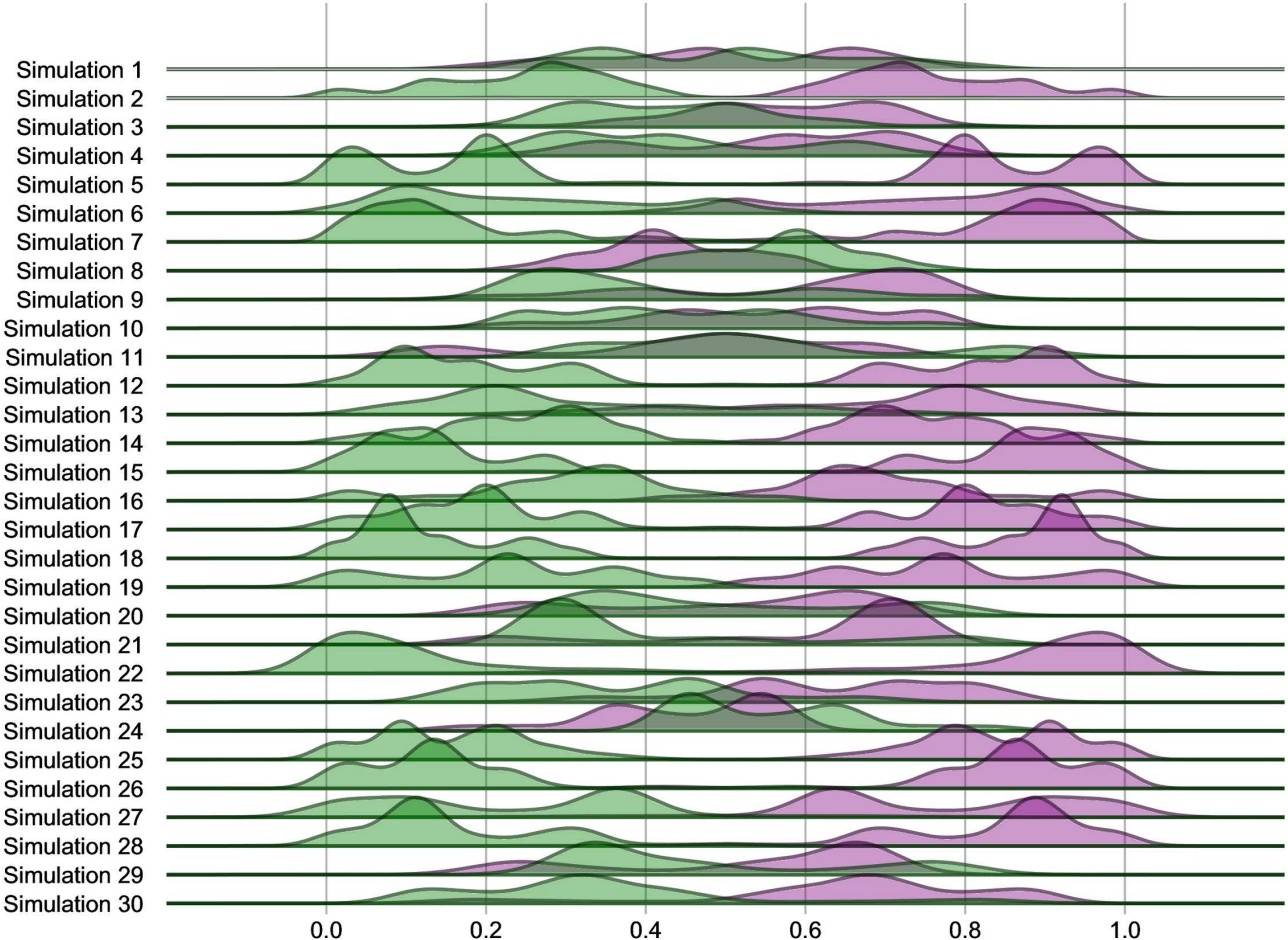

**Fig 3. Ridgeline plot showing the distributions of proportions of infected P and M females.** The distributions of proportions of infected P and M females (in purple and green, respectively) were realised using the values generated at each tick in 30 simulations, from tick 10 to the last tick in which coexistence between infected P and M females was observed, and excluding runs where the index cases were all randomly assigned to a single strategy. This scenario uses the baseline condition settings, 10 *Index-Cases*, null *Offspring-Reduction-%* determined by the infection (0%), and 100% *Probability-of-Transmission-%* per sexual contact in the context of a SI system without vertical transmission of the disease.

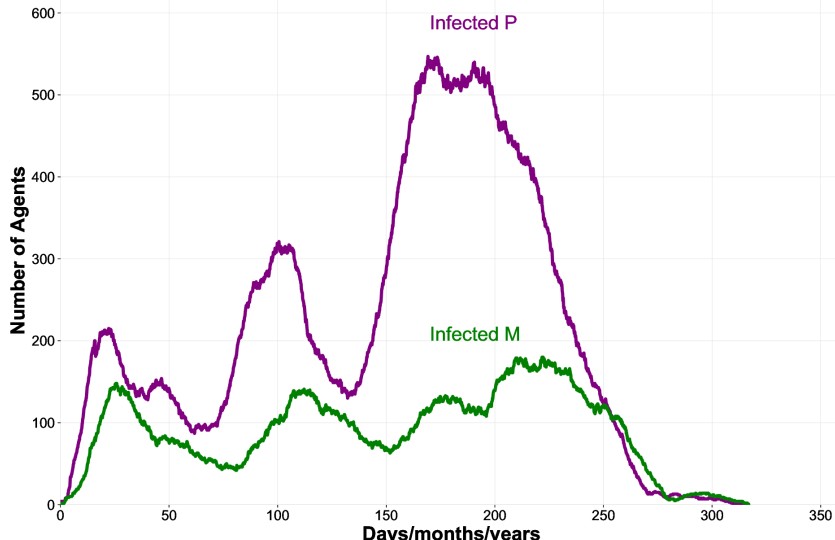

**Fig 4. Comparison of the number of females affected by the infection for each mating strategy (example from a single run).** Numbers of infected P females (in purple) and infected M females (in green) show the prevalence of the disease among polyandrous strategists. We introduced the pathogen in the context of the baseline condition (i.e., 100 *Initial-Males*, 50 *Initial-Mfemales*, 50 *Initial-Pfemales*, *Juvenile-Threshold* at 1500, *Lifespan* at 150, *Age-of-Sexual-Maturity* at 10, *Maximum-Matings* at 5, and *Time-Out-of-the-Market* at 6) and set 10 *Index-Cases*, null *Offspring-Reduction-%* determined by the infection (0%), and 100% *Probability-of-Transmission-%* per sexual contact, using the SI version of the model and excluding vertical transmission of the pathogen.

multiple matings expose more easily those females to contact with infected males. Conversely, the persistence of the pathogen in a homogeneous M population over time is quite modest. It should be noted that these results on the epidemic dynamics were obtained in scenarios without costs for infection (because the infectious agent is "neutral"). So, we can conclude that, considering the baseline condition, polyandry is a catalyser for the infection, and P females are the major carriers of the pathogen. This is another basic finding that, from a model validation perspective, supports the reliability of the outcomes generated by the scenarios contemplated in this study and discussed below, which are all characterised by the mutual interaction between mating systems and the STD. Indeed, simulations are able to capture the fundamental dynamics which are commonly observed in systems analogous to those considered in this work [32,34,36,78,110,111].

To further explore the interplay between mating strategies and epidemic dynamics, medium-high and high probabilities of transmission (i.e., 70%, 85%, and 100%) were tested under SA conditions considering both a system with horizontal transmission of the infection and a system with horizontal and vertical transmission (i.e., from mothers to their offspring) (Fig 5). In these simulations we also applied an infection cost (*Offspring-Reduction-%*) of 1%. As we can see from the system including exclusively horizontal transmission (Fig 5A and Fig 5B), under equal costs and benefits of mating (that is, when polyandry is neither beneficial nor costly in the absence of infection), the higher the transmissibility of the pathogen the less likely the P strategy ultimately reached fixation. Interestingly, when vertical transmission was introduced into the system (Fig 5C and Fig 5D), everything else being equal, the relative frequency of fixation of the M strategy was always close to 100% regardless of the probability of transmission. This finding suggests that vertical transmission itself may be a critical factor which can decrease the probability of fixation of the P strategy drastically. Moreover, it may be equally impactful regardless of the ability of the pathogen to spread horizontally.

In a nutshell, these outcomes suggest that the probability of transmission of a STD and the fixation rates of the M genotype/phenotype/strategy may be positively related when considering an epidemiological context which is exclusively based on horizontal transmission of the infection (Fig 5A). Conversely, mean times to fixation seem to be more uniform (Fig 5B).

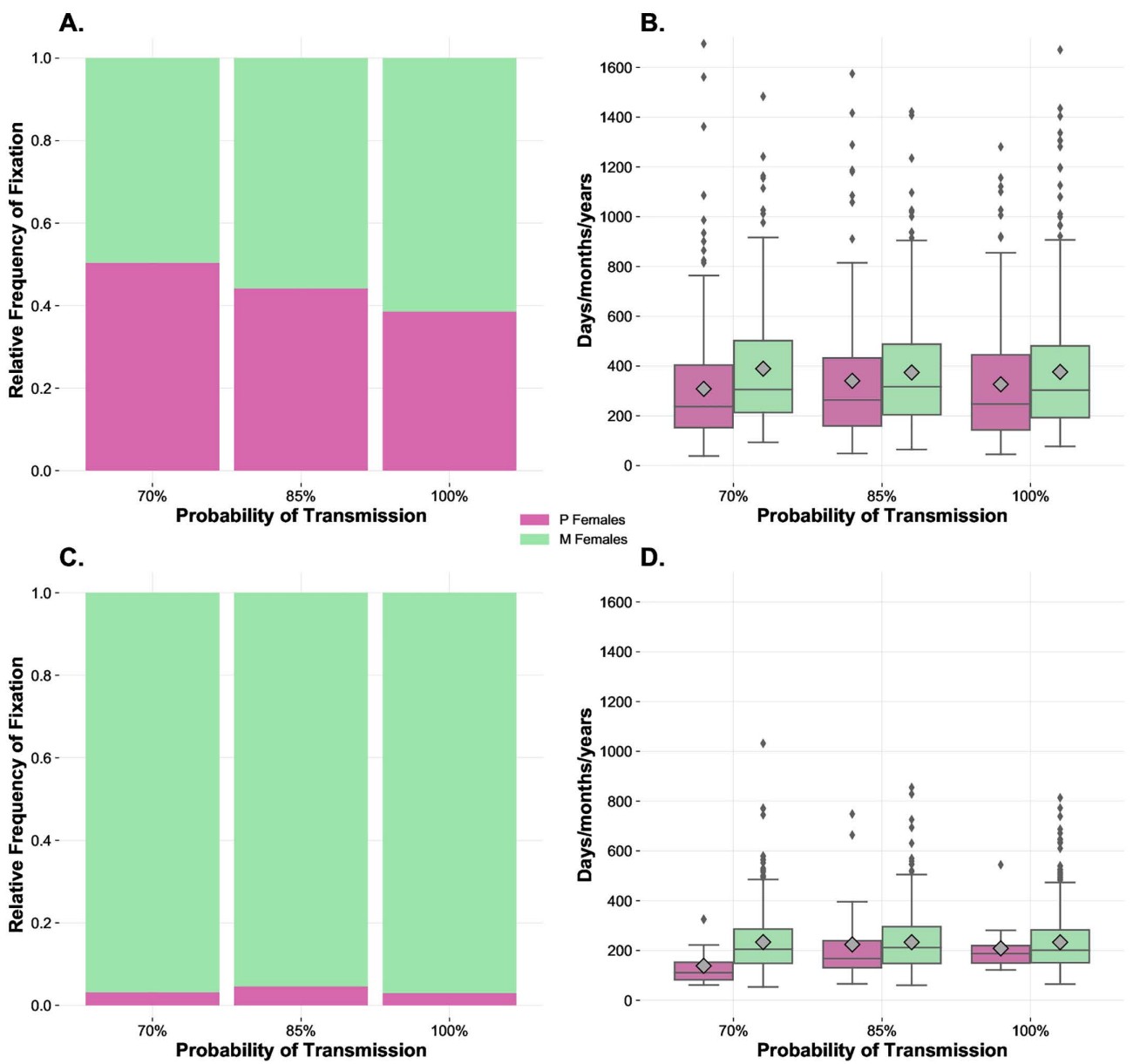

**Fig 5. The effects of variation in the probability of transmission per sexual contact on the relative frequency of fixation and time to fixation of each strategy.** This SA involves 200 initial individuals, 1:1 sex ratio, even distribution of genotypes/mating strategies, fixed *Benefit-of-Mating-%* and *Cost-of-Mating-%* at 3%, 10 *Index-Cases*, 1% *Offspring-Reduction-%* due to the infection, and variable *Probability-of-Transmission-%* of the pathogen (70%, 85%, and 100%). Panels **A** and **B** show the results when vertical transmission of the pathogen is not allowed, whereas panels **C** and **D** show the results when both modalities of transmission, that is, horizontal and vertical, occur. Both alternatives were simulated within a SI system. Relative frequency of fixation values indicate the number of simulations where each genotype reached fixation out of the total number of simulations (500) **(A, C)**, while Mean time to fixation values **(B, D)** indicate the mean time each genotype took to reach fixation and are represented by the grey symbols inside the boxplots. Paired boxplots provide information on the distributions of the times to fixation obtained.

The fact that the transmissibility of the pathogen is a relevant variable within a system where vertical transmission of the disease is absent has also been hinted by previous research [30,112]. Notably, our results emphasise that the pathogen transmissibility is capable of moderating mating system evolution even in the absence of differences in infection-induced

costs. However, at least when considering a SI system, vertical transmission could reduce the impact of the probability of transmission per sexual contact as a variable capable of modulating the evolution of mating strategies, which indicates the importance of the interaction between the many factors at play.

### Sensitivity of the system to variations in the probability of transmission

One of our main objectives with this study was to analyse the response of our computational system to variations in the probability of transmission of the pathogen in several alternative contexts. In the following set of simulations, we kept track of both the frequency of cases of endemization (i.e., stable but low circulation of the pathogen over time), and the probability of transmission that can induce this steady state in the STD dynamics (Fig 6). For this simulation scenario, we inspected a homogeneous P population and considered several probabilities of transmission in the context of two alternative epidemiological systems, that is, a SI and a SIS system, respectively. In the case of a SI system, the results indicated a clear tipping point where a change from moderate to medium-high probabilities of transmission (60% to 70%) led to a much greater stability of the infection over time, to the extent that, with a probability of transmission of 70%, all 100 simulations showed the persistence of the STD for over 1000 days/months/years (depending on the unit of time ascribed to each 10 ticks). Conversely, such persistence of the pathogen over time was never observed, not even in a single simulation, when transmission rates below 70% were considered. The same behaviour was observed using a SIS system and a moderate value of *Time-to-Recover* (= 50 ticks), with the only difference being a shift of the tipping point leading to endemic states from a probability of transmission of 70% to 90%. In fact, when considering transmission rates below 90%, the persistence of the STD was always very limited in time, whereas the infection became stable for more than 1000

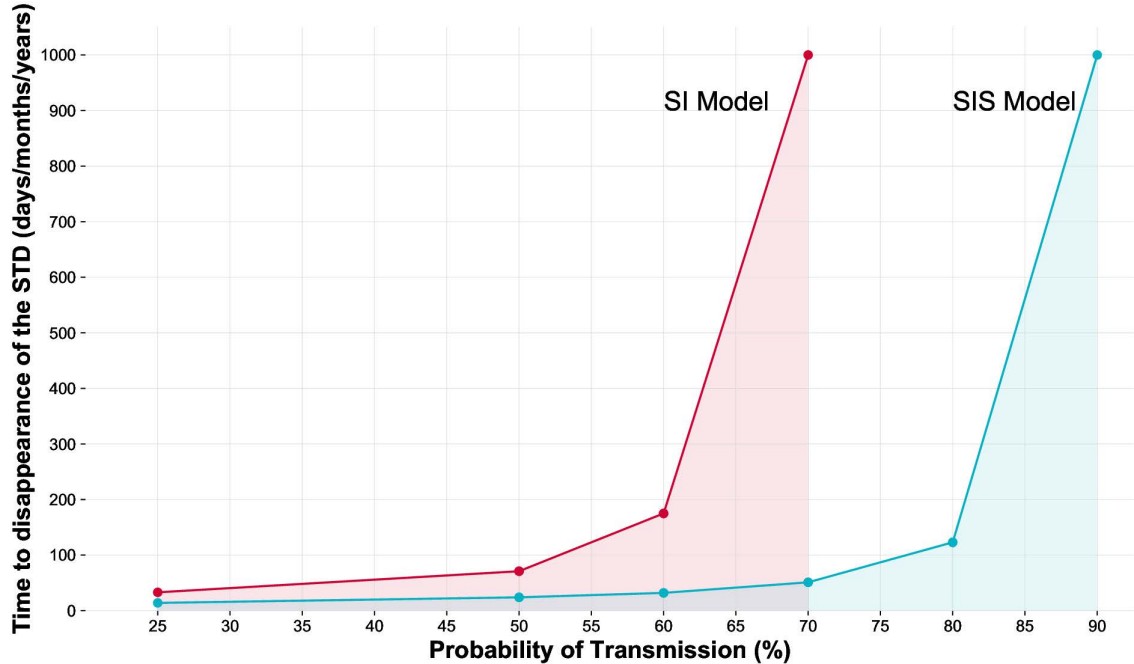

**Fig 6. The effects of variation in probabilities of transmission per sexual contact on persistence of the epidemic over time (endemization) in a SI and a SIS system.** This analysis was carried out using 100 *Initial-Males*, 100 *Initial-Pfemales* (i.e., monogamous phenotype is absent), *Cost-of-Mating-%* and *Benefit-of-Mating-%* fixed at 3%, 10 *Index-Cases*, 1% *Offspring-Reduction-%* due to the infection, a *Time-to-Recover* of 50 ticks in the SIS model, and variable *Probability-of-Transmission-%* of the pathogen (25%, 50%, 60%, and 70% using a SI model, 25%, 50%, 60%, 70%, 80%, and 90% using a SIS model). Simulations were stopped at ten thousand ticks (i.e., 1000 days/months/years, depending on the system envisaged; see text) if extinction of the pathogen had not occurred by then.

days/months/years in 99/100 runs with a probability of transmission of 90%. The distributions of the time elapsed until the disappearance of the pathogen across all 100 runs of each setting that did not lead to endemization, that is, using probabilities of transmission below the endemicity thresholds detected here, are shown in S1 Fig (SI model) and S2 Fig (SIS model).

The existence of an endemic steady state characterised by a low, yet stable persistence of the pathogen over time when using a probability of transmission per contact of 70% in the SI model was also supported by additional simulations (see Fig 7). In addition, the distributions of proportions of infected individuals, which were obtained by using the latter settings and extracting values of proportion at each tick from tick 10 (i.e., the age of sexual maturity) up to tick 10000 in 30 runs (S3 Fig) seem to indicate the presence of a moderate-low percentage of infected agents during simulations, predominantly between 10% and 15%.

A relevant aspect of our study involving non-linearity concerns the emergence of these endemic steady states. We recall that the main peculiarities of the model we used to obtain the outcomes described above are the following: i) the presence of a sexually transmitted disease with variable rate of transmission, ii) a fully polyandrous context, iii) panmictic mating, iv) male polygyny, v) a randomly selected degree of polyandry per female, and vi) a refractory period to further copulations which is applied to each female after mating. Therefore, the relevance of these results depends on the information they provide on the specific ecological context to which they refer, indicating that a medium-high or high transmissibility per sexual contact would make a sexual infection difficult to be eradicated in all cases where the assumptions listed above apply. Nevertheless, this limitation would not prevent us from finding thresholds very close to those we found here in many animal case studies involving a wide variety of diseases (see Fig 1). To the best of our knowledge, the thresholds presented here are two of just a few cases in which the existence of such tipping points underlying endemization and transmissibility have been found in the context of mating strategies and disease [35,39], although critical transmission rates and specific values of reproductive ratios ($R_0$) that represent epidemic thresholds beyond which epidemics spread, and below which they do not occur, are well known in epidemiology [113,114]. Considering the concept of $R_0$, our findings could actually represent two alternative situations where $R_0 = 1$ and the spread of the pathogen tends to be stable over time, so their significance would lie in their applicability to the specific epidemiological context examined here, which has certain characteristics that may not be easily found in human systems.

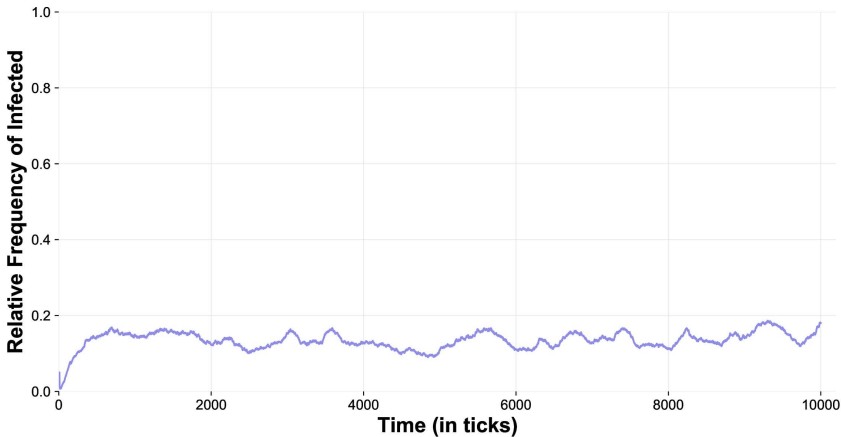

**Fig 7. Curve of infected individuals (relative frequency) obtained with a 70% probability of transmission in a SI system.** This test involved 200 individuals, 1:1 sex ratio, 100 *Initial-Pfemales* (i.e., there are no monogamous strategists), *Cost-of-Mating-%* and *Benefit-of-Mating-%* fixed at 3%, 10 *Index-Cases*, 1% *Offspring-Reduction-%* due to the infection, and 70% *Probability-of-Transmission-%* per sexual contact. In this case, we kept track of the relative frequency of infected agents for each of the 10000 ticks of a single, representative simulation and obtained the mean value 0.134 (SD = 0.023). This plot shows the attainment of endemization in this scenario.

This scenario, particularly in its SI version, was also highly relevant for testing the maintenance of the endemic thresholds detected here across different ecological conditions, that is, when modifying the values assigned to some fundamental variables of our model, which were instead kept constant in all other scenarios. The additional experiments we performed to carry out this further exploration included settings identical to those used to identify the endemicity threshold in the SI model, except for the use of a fixed value of *Probability-of-Transmission-%* (70%, which always allowed endemization in the experiments shown above) and different values of *Juvenile-Threshold* and *Time-Out-of-the-Market*, respectively.

Testing the sensitivity of the system to variations in the upper limit of coexisting juvenile individuals by setting *Juvenile-Threshold* at 500, 1000, 1500 (our default value), and 2000 informed that population size and random mortality of younger animals may influence significantly the outcomes. More specifically, while values of *Juvenile-Threshold* ≥ 1000 always guaranteed reaching of the endemic state in all 100 runs executed for each setting, *Juvenile-Threshold* = 500 enabled reaching of the endemic state in 74 runs out of 100, confirming the full validity of our finding only above a certain critical number of juvenile agents.

In contrast, consistently with our preliminary analyses, we found that neither reducing the default value we used for *Time-Out-of-the-Market* (6 ticks) by two-thirds (2 ticks) nor tripling it (18 ticks) prevented reaching a stable pathogen circulation over time, which still occurred in all runs. Surprisingly, a relevant reason why a 70% probability of transmission seems to be sufficient to guarantee endemization across the three different conditions explored in this additional test may lie in the fact that the total number of sexual contacts recorded at the end of every run, i.e., after 10000 ticks (1000 days/ months/years) was remarkably stable around 269000. More specifically, using a refractory period of 2 ticks, the mean number of sexual contacts across runs was 269553.51 (SD = 756), while this average value was 269137.91 (SD = 602.39) and 269673.29 (SD = 651.07) when using refractory periods of 6 and 18 ticks, respectively. Thus, changing the rate of production of new batches of offspring by modifying *Time-Out-of-the-Market* (in the model, oviposition by P females always occurs at the same time as mating) might neither reduce nor increase the number of contacts significantly, which could make the duration of P females' refractoriness to new matings irrelevant in this context. This suggests that the endemization threshold detected here is potentially shared by many animal systems with refractory periods of different magnitudes.

### The disappearance of the epidemic as a predictor of the extinction of polyandry

In a subsequent experimental scenario ("*Co-occurrences of the extinction of the pathogen and loss of the P phenotype*"), we followed tick after tick the evolution of the system, focusing on the concurrent P-pathogen extinction (that is, when both the epidemics and the allele-inducing polyandrous behaviour ultimately disappear from the population), considering very high transmissibility (100%). We simulated a Symmetric Analysis (SA), where the mating strategies were evenly distributed at the beginning of each simulation. In 196 out of 250 simulations both the infection and the P strategy faced the same outcome, that is, either disappearance from the population or stable coexistence. In the 54 remaining simulations, the pathogen disappeared while the P strategy persisted for a longer time. As for the cases where both the P strategy and the pathogen reached extinction (i.e., 118 runs), in 116/118 simulations the STD disappeared first, and the P strategy followed several ticks after. In other words, in most cases, the end of the epidemics preceded the extinction of polyandry. The correlation coefficient between time to disappearance of the pathogen and time to disappearance of P females across simulations (n = 118) is $r = 0.688$ ($p = 7.07E-18$, see Fig 8 to visualise the relationship between these two variables).

This scenario brings at least two different pieces of information on the evolution of the system: (i) the fixation of the P strategy leads to the endemization of the epidemics, and (ii) when the pathogen disappears, the extinction of P frequently follows soon after. Given the previous outcomes, the latter behaviour of the system may seem counterintuitive, since the P strategy should be favoured by the disappearance of the pathogen and eventually reach fixation. Nevertheless, a possible explanation for this phenomenon is the advantageous condition that the M strategy "inherits" from the extinguished epidemic: the infection is the only disruptive force considered in this scenario and, although at some point it disappears, it

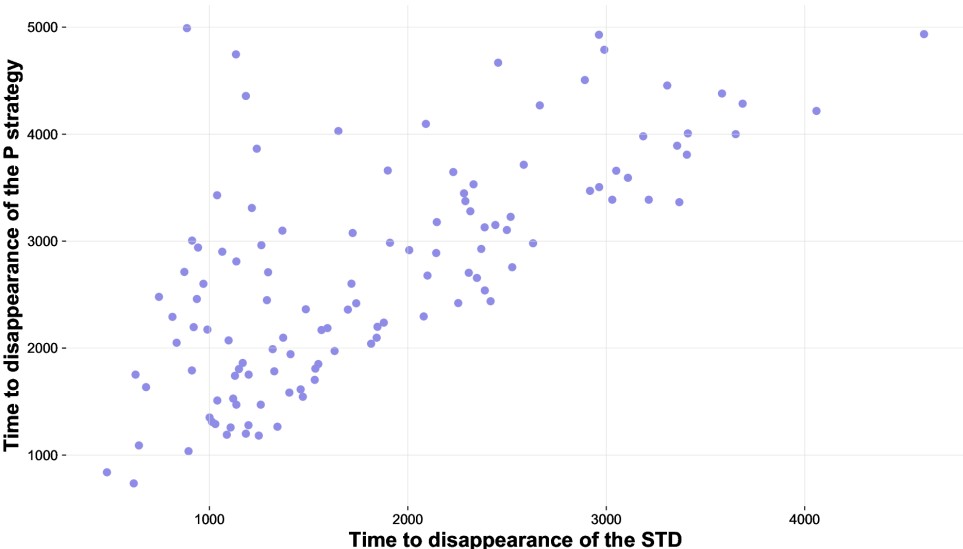

**Fig 8. Relationship between the time to disappearance of infected individuals and the time to disappearance of polyandrous females.** This scatter plot shows the relationship between the time to disappearance of infected individuals and the time to disappearance of the polyandrous strategy, only considering simulations where both infected agents and polyandrous females ultimately disappeared from the population.

could make the two strategies restart from a radically different situation in terms of frequency. In other words, given a context characterised by equal *Cost-of-Mating-%* and *Benefits-of-Mating-%* (if all individuals are healthy), the infection-related cost *Offspring-Reduction-%* could reduce the number of P females to the point that, when the outbreak ends, it could be almost impossible for them to regain a wide presence in the system, even if they are not penalised anymore. Several lines of evidence support this explanation for the observed pattern, including the strong association between the infection and the polyandrous strategy, which was discussed above.

Furthermore, that the recovery of P frequency is unlikely when the proportion of P females in the system is low can be illustrated carrying out an additional experimental scenario in which, for instance, 200 *Initial-Males*, 150 *Initial-Mfemales* and 50 *Initial-Pfemales* (that is, a moderate, conservative number of P females), compete under no infection and equal *Benefit-of-Mating-%* and *Cost-of-Mating-%*. Under these conditions, as expected, the P strategy reached fixation in only 22 times out of 100, which suggests an appreciable disadvantage induced by a sizeable, yet not excessive, numerical inferiority. The relationship between probability of fixation of each strategy and its initial frequency (when costs and benefits of mating are equal) was also supported by a second additional scenario, this time with a more pronounced numerical difference between the two strategies (but always with the same initial population size). This scenario was identical to the latter, except for the fact that it involved 200 *Initial-Males*, 190 *Initial-Mfemales*, and 10 *Initial-Pfemales*. In this case, P strategy reached fixation in only 5 out of 100 total simulations. These results obtained under the baseline conditions in our system are obvious, but they serve to illustrate that the lower the number of P strategists, the more difficult it is to succeed in prevailing over M females, even when polyandry is not costly. The basic mechanism underlying these additional outcomes, which relies on the higher vulnerability of small proportions of P females to random fluctuations, is a possible, although probably not exclusive explanation of the counterintuitive information provided by this "Co-occurrences" scenario, which clearly shows that the disappearance of the infection is almost always followed by a rapid extinction of all P strategists. If this justification is valid as we believe, then stochasticity affecting a reduced pool of polyandry-related alleles could be a main dynamic governing this phenomenon. However, this unexpected pattern might be influenced by multiple factors. Indeed, genetic drift alone (in the absence of the infection, alleles associated with P and M would be neutral in this

scenario) would hardly explain the earlier disappearance of the pathogen compared to that of the P strategy. The pattern of pathogen first-P strategy second sequential disappearance is potentially due to the difficulty of the infection to spread when the composition of the population in terms of mating strategies makes transmission of the pathogen too unlikely. Regardless of the completeness of the explanation provided here, the "order of precedence" uncovered by our computational system could be relevant for practical aims, as it highlights the possible role of the end of an epidemic as a sufficiently reliable predictor of the prevalence of monandrous strategies in an ecological niche or, to put it another way, as a potential signal of a doomed promiscuous strategy.

## Medium-low transmissibility reveals a switch-like behaviour when polyandry is highly beneficial

In the SA scenario named "*High benefits, variable transmissibility*", we inspected the dynamics of the infection when the pathogen entered a population characterised by very high *Benefit-of-Mating-%* (5%) and medium-high *Cost-of-Mating-%* (4%), that is, promoting polyandrous behaviour in the absence of infection. Several probabilities of transmission (i.e., 25%, 50%, 75%, and 100%) were tested within an epidemiological context characterised by recovery after infection and both horizontal and vertical transmission of the disease (SIS system with *Time-to-Recover* = 50 ticks). Also in this case, a turning point was detected when considering a probability of transmission between 25% and 50%, which reveals a non-linear relationship between the probability of transmission and the frequency of fixation of the M strategy. Indeed, while a low transmission rate was not sufficient to enable a significant frequency of fixation of the M strategy (7.2% of runs), setting a probability of transmission of 50% allowed the M strategy to reach fixation in almost one case out of five (16.2% of runs), which was slightly below the frequencies of fixation of M observed when using a probability of transmission equal to 75% and 100% (i.e., 17.8% and 19.2% of runs, respectively) (Fig 9).

Interestingly, transmissibility proves again to be a critical variable in mating systems when considering a SIS context also including vertical transmission of the disease. Indeed, when mating entails substantial *Benefit-of-Mating-%* (5%, with *Cost-of-Mating-%* of 4%), the system shows a sudden, although moderate switch-like behaviour when considering a value of *Probability-of-Transmission-%* between 25% and 50%, with a clear increase in the advantage of the M strategy compared with the lowest probability value used (25%). This acceleration, or "phase transition", borrowing from the field of physics, shows well how unpredictable emerging patterns generated by the complex interactions involved in a mating system may be, especially if the system is placed within the context of a STD. Furthermore, these simulations are also of biological interest, aside complex systems theory. Indeed, even in a context where polyandry is highly favoured (e.g., thanks to nutritive ejaculates, nuptial gifts, etc.), a STD with both horizontal and vertical transmission and a medium-low transmissibility is perfectly able to reduce those initial advantages for the P phenotype. The simultaneous implementation of general benefits and costs of mating and costs induced by the disease allowed our model to add further critical details to the knowledge gathered in previous studies dealing with the effects of different transmissibility rates in heterogeneous mating systems [27].

As for the costs and benefits induced by polyandrous behaviour or promiscuity [1], our findings may also improve the understanding of the relationship between fitness-reducing STDs and polyandry as a bet-hedging strategy. By mating with multiple mating, polyandrous females may be considered to be "hedging their bets", either because they increase the genetic diversity among their offspring (useful under fluctuating and unpredictable environmental conditions), or because a broader sampling of males in the population minimizes the risk of complete reproductive failure, e.g., due to male infertility problems or genetic parental incompatibilities [104,115]. The advantages that bet-hedging females gain by mating multiply may be an important reason why polyandry has been documented to persist within populations even in risky conditions, more specifically, in presence of STDs [31]. Our study can provide some insights into these questions, for instance, by highlighting the conditions where infection-related risks lessen the benefits of polyandry and, possibly, of a bet-hedging strategy. Indeed, since polyandry is the main vehicle for the transmission of the pathogen in our system, the finding that medium-low probabilities of transmission may undermine even extensive benefits of polyandry suggests some possible

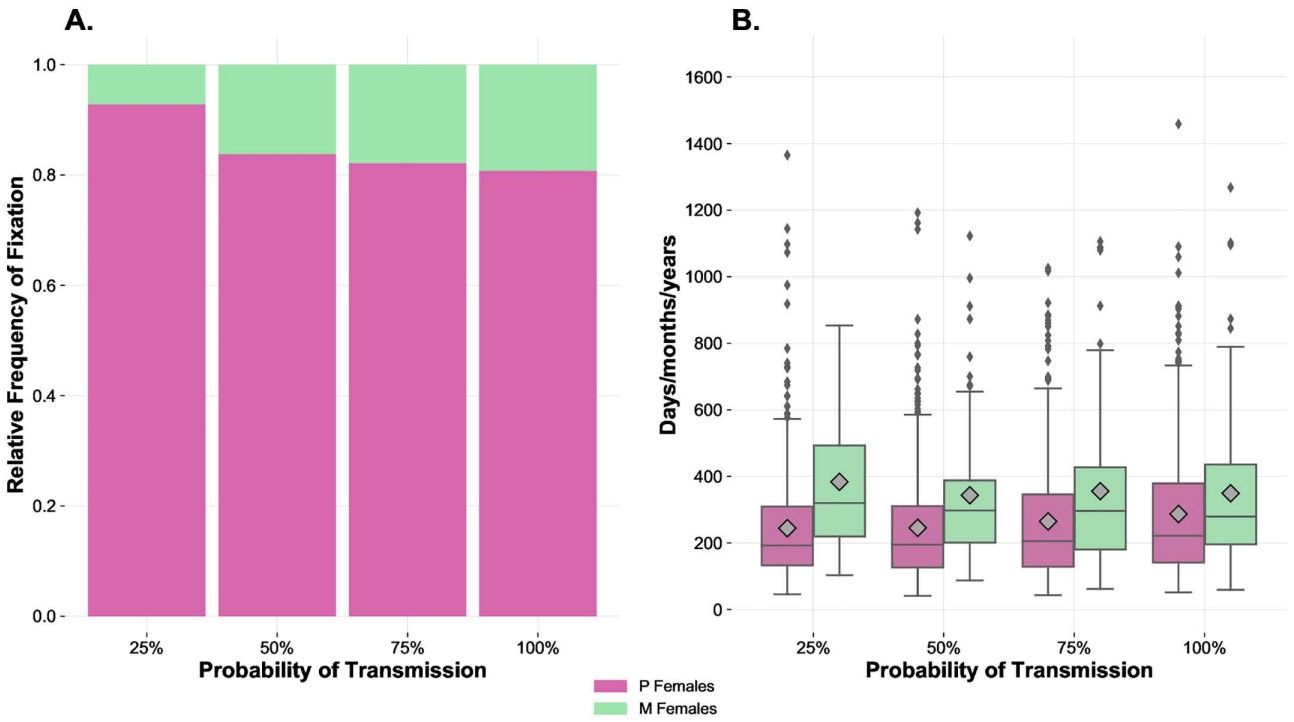

**Fig 9. Effects of variation in probabilities of transmission per sexual contact on the relative frequency of fixation and time to fixation in a context of very high direct benefits of mating, with recovery after infection and vertical transmission.** This SA was carried out using 200 initial individuals, 1:1 sex ratio, even distribution of the two strategies, fixed *Benefit-of-Mating-%* at 5%, fixed *Cost-of-Mating-%* at 4%, 10 *Index-Cases*, 1% *Offspring-Reduction-%* due to the infection, *Time-to-Recover* = 50 ticks (SIS system), and variable *Probability-of-Transmission-%* of the pathogen (25%, 50%, 75%, and 100%). **(A)** Relative frequency of fixation values indicate the number of simulations where each genotype reached fixation out of the total number of simulations (500). **(B)** Mean time to fixation values indicate the mean time each genotype took to reach fixation and are represented by the grey symbols inside the boxplots. Paired boxplots provide information on the distributions of the times to fixation obtained.

boundaries for bet-hedging benefit returns and also indicates that mating systems are sensitive to small variation in key factors.

### Effects of variable transmissibility when the cost-benefit gap is large

We were also interested in testing the effects of extreme conditions in terms of difference between costs and benefits of mating. The last scenario we simulated was an IA ("*Variable transmissibility, large cost-benefit gap*") considering the SIS version of the model with both horizontal and vertical transmission of the infection, a *Benefit-of-Mating-%* of 2%, a *Cost-of-Mating-%* of 5% and a single M mutant/immigrant. Also in this case, we used a *Time-to-Recover* of 50 ticks (i.e., 5 days/months/years) and several values of *Probability-of-Transmission-%* (25%, 50%, 75%, 100%). While the very high cost of mating allowed the M strategy to reach fixation in several runs, changing the probability of transmission had no relevant effect on the relative frequency (Fig 10A) and mean time to fixation (Fig 10B) of both strategies. Keeping in mind the importance of probabilities of transmission in the previous scenario, which was also about a SIS system with vertical transmission of the pathogen, the irrelevance of this variable in this IA is another unexpected outcome that might be explained by a reduced impact of STDs when the cost of mating is so high as to significantly penalise even those females which mate with only one partner during their lifetime.

The last unexpected behaviour we focus on, this time obtained by using a SIS model with vertical transmission of the infection and a large gap between costs and benefits of mating, is the irrelevance of changing probabilities of transmission

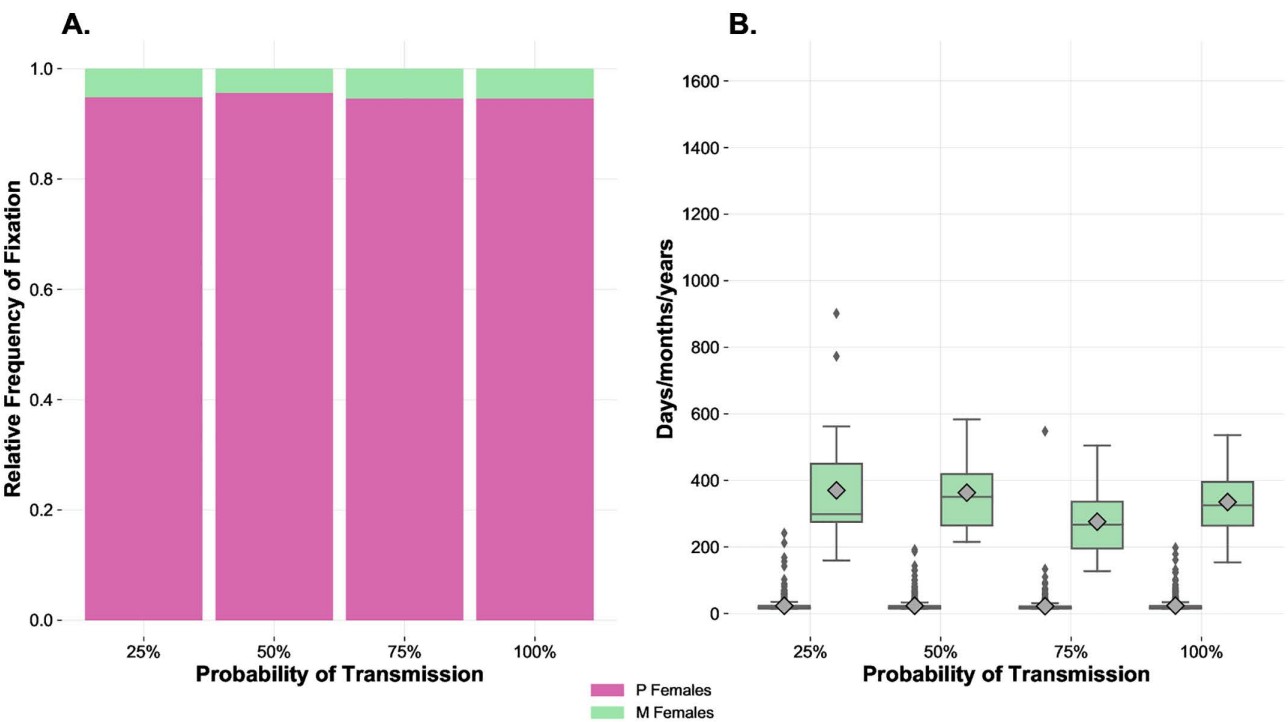

**Fig 10. Effects of variation in the probability of transmission per sexual contact on the relative frequencies of fixation and mean times to fixation in the context of a very large gap between costs and benefits of mating, with recovery after infection and vertical transmission.** This IA was carried out using 200 initial individuals, 1:1 sex ratio, 100 *Initial-Males*, 99 *Initial-Pfemales*, 1 *Initial-Mfemales*, fixed *Benefit-of-Mating-%* at 2%, fixed *Cost-of-Mating-%* at 5%, 10 *Index-Cases*, 1% *Offspring-Reduction-%* due to the infection, *Time-to-Recover* = 50 ticks (SIS system), and variable *Probability-of-Transmission-%* of the pathogen (25%, 50%, 75%, and 100%). **(A)** Relative frequency of fixation values indicate the number of simulations where each genotype reached fixation out of the total number of simulations (500). **(B)** Mean time to fixation values indicate the mean time each genotype took to reach fixation and are represented by the grey symbols inside the boxplots. Paired boxplots provide information on the distributions of the times to fixation obtained.

of the pathogen on the mating system. Indeed, this IA may represent a case in which the STD is not a factor that is capable of influencing in a relevant way the global dynamics, which are instead generated by the costs and benefits associated with mating. We suspect that, while negatively affecting the reproductive performances of polyandrous females, relevant costs of mating might also limit the emergence of alternative mating strategies, at least when the initial frequency of the invader strategy is very low. From an evolutionary point of view, this could mean that even in epidemiological contexts where a bet-hedging multiple mating strategy brings very limited advantages, the extinction of the polyandrous strategy might not be the most likely outcome if mating itself is very costly. And this would be the case even if the STD is highly transmissible. Indeed, the high cost of mating might hinder a far greater advantage that would otherwise be conferred on M females by the epidemics. However, we feel that this kind of phenomena would deserve further investigation in future assessments.

## General considerations on the agent-based model

More generally, our agent-based modelling approach to the study of the interrelationship between mating strategies and epidemiological dynamics informs that this interplay may be remarkably complex and may have important repercussions for both the spread of disease and the evolution of sexual behaviours and mating systems. Our model shares some elements with previous models investigating the reciprocal influences between mating system and STDs but incorporates

new angles. For example, Theuer and Berec [112]'s study investigated mate choice based on infection avoidance, and, as our study, considered a Susceptible/Infected system with sexual transmission, balanced sex ratio, variance in the probabilities of transmission, and fecundity costs for infected individuals, but it did not consider costs and benefits of mating other than those related to the infection. Similarly, in McLeod and Day [27]'s model, different costs and benefits, other than the cost inflicted by the disease, were not contemplated. An important feature of our approach is that it, as other studies have done previously [29,32], implemented variation in female mating rates, a realistic feature of natural populations. Furthermore, by considering a "Time out of the mating pool" variable [41,42,116], our model contained a further element of realism, analogously to some network-based models where partners are not available for sexual interactions all the time [e.g., 37].

## Conclusions

In summary, our results collectively support the notion that the evolution and maintenance of monogamous or polygamous mating systems and strategies are tightly associated to epidemic contexts, and vice versa. Our agent-based modelling approach generated results that support findings in previous studies, but it also offered new far-reaching insights. For instance, it revealed that the disappearance of the infection under some circumstances may serve as an indicator of the reduction of heterogeneity regarding mating strategies in a population, and, ultimately, of the extinction of female multiple mating genotypes. Our modelling approach was specifically designed to detect whether the system would show non-linear outcomes and unexpected patterns emerging from multiple interactions, both between the two mating system strategies, as well as between the mating system and the STD. Indeed, in two of our experiments, we found some remarkable tipping points changing the overall behaviour of the system, sometimes even dramatically: medium-low probabilities of transmission were found to be critical to reduce the advantage of polyandrous females with even very high benefits of mating, whereas medium-high or high probabilities of transmission were found to induce endemization of the infection.

Our results collectively demonstrate that investigating the reciprocal interactions between disease spread dynamics and sexual behaviour can offer valuable insights into disease transmission patterns and the evolution of reproductive strategies in both wildlife and human populations. In human populations, our results underscore the importance of targeted education campaigns, promoting safe-sex and regular testing, focusing on individuals who engage with multiple sexual partners, to significantly reduce transmission rates. Our study also highlights the utility of knowledge concerning mating systems and sexual behaviour in wild or domestic animals for assessing STD propagation. Integrating behavioural insights into disease management strategies and STD prevention can lead to more effective interventions across diverse contexts, including conservation biology of wild animals, biological control of pests, or animal production.

## Supporting information

**S1 Appendix. NetLogo model.** The agent-based model can be opened by installing the NetLogo simulation platform (version 6.2.0), which can be downloaded free of charge from the official website of the software at the following link: https://ccl.northwestern.edu/netlogo/6.2.0/.
(NLOGO)

**S1 Fig. Ridgeline plot showing the distributions of the time elapsed until the disappearance of the pathogen in the SI version of the model.** The plot reports the distributions of the time (in ticks) elapsed until the disappearance of the pathogen across all 100 runs of each setting tested in the scenario named "*Minimum probability of transmission for endemization*" which did not lead to endemization in a SI system (i.e., that involved probabilities of transmission < 70%).
(TIF)

**S2 Fig. Ridgeline plot showing the distributions of the time elapsed until the disappearance of the pathogen in the SIS version of the model.** The plot reports the distributions of the time (in ticks) elapsed until the disappearance of

the pathogen across all 100 runs of each setting tested in the scenario named "*Minimum probability of transmission for endemization*" which did not lead to endemization in a SIS system (i.e., that involved probabilities of transmission < 90%). (TIF)

**S3 Fig. Ridgeline plot showing the distributions of proportions of infected individuals.** The plot reports the distributions of proportions of infected agents in the SI version of the scenario named "*Minimum probability of transmission for endemization*", using the minimum probability of transmission needed to reach an endemic state in all runs (i.e., 70%). The proportions were extracted at each tick of 30 simulations, from tick 10 (the age of sexual maturity in our scenarios) up to tick 10000.
(TIF)

## Acknowledgments

We wish to thank Paula Martin Art for her valuable suggestions that allowed us to improve the Figs presented in this article.

## Author contributions

**Conceptualization:** Riccardo Tarantino, Francisco Garcia-Gonzalez.

**Data curation:** Riccardo Tarantino.

**Formal analysis:** Riccardo Tarantino.

**Funding acquisition:** Francisco Garcia-Gonzalez.

**Investigation:** Riccardo Tarantino, Francisco Garcia-Gonzalez.

**Methodology:** Riccardo Tarantino, Francisco Garcia-Gonzalez.

**Software:** Riccardo Tarantino.

**Supervision:** Francisco Garcia-Gonzalez.

**Validation:** Riccardo Tarantino, Francisco Garcia-Gonzalez.

**Visualization:** Riccardo Tarantino.

**Writing – original draft:** Riccardo Tarantino, Francisco Garcia-Gonzalez.

**Writing – review & editing:** Riccardo Tarantino, Francisco Garcia-Gonzalez.

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
