## [Decision Letter · Decision Letter 0]

27 Dec 2024

Dear Dr. Tarantino,

Thank you for submitting your manuscript to PLOS ONE. After careful consideration, we feel that it has merit but does not fully meet PLOS ONE’s publication criteria as it currently stands. Therefore, we invite you to submit a revised version of the manuscript that addresses the points raised during the review process.

We look forward to receiving your revised manuscript.

Kind regards,

Aurelio A. de los Reyes V, Dr. rer. nat.

Academic Editor

PLOS ONE

Journal Requirements:

3. Please include a caption for figure 6.

4. We are unable to open your Supporting Information file S1_Appendix.nlogo. Please kindly revise as necessary and re-upload.

5. We note you have included a table to which you do not refer in the text of your manuscript. Please ensure that you refer to Table 2B in your text; if accepted, production will need this reference to link the reader to the Table.

Reviewers' comments:

Reviewer's Responses to Questions

**Comments to the Author**

1. Is the manuscript technically sound, and do the data support the conclusions?

Reviewer #1: Yes

Reviewer #2: No

2. Has the statistical analysis been performed appropriately and rigorously?

Reviewer #1: N/A

Reviewer #2: N/A

3. Have the authors made all data underlying the findings in their manuscript fully available?

Reviewer #1: Yes

Reviewer #2: Yes

4. Is the manuscript presented in an intelligible fashion and written in standard English?

Reviewer #1: Yes

Reviewer #2: Yes

Reviewer #1: Dear authors,

I would like to thank the authors for their submission. The study provides some new insights regarding the relationship between mating system evolution and sexually transmitted diseases. I think the study has some merits for publication, provided the authors can clarify and address the issues outlined below, particularly regarding the assumptions made in agent-based model, which are crucial for trusting the implications derived from it. Additionally, it would be helpful to discuss the major limitations of the study, allowing readers to more effectively assess the results.

Major comments

1. Could you clarify which type of epidemic model you adopted (e.g., SI, SIS)? Additionally, it would be helpful to include a discussion about the potential implications of considering other "classes of agents," such as recovery, especially given that some STDs can confer temporary immunity from infection. The choice of model structure is crucial, as it can significantly influence the dynamics and outcomes of disease epidemiology. It might also be worth mentioning this as a limitation if recovery or immunity was not incorporated into the model.

2. Vertical transmission is an important component of transmission models, particularly in the context of STDs. It would be valuable to include a discussion on its potential impact on the model's outcomes and dynamics, as it could significantly influence the overall disease trajectory and population-level effects.

3. It would be helpful to provide a more detailed description of the randomization process in your agent-based model (e.g., random number generator used (e.g., uniform, Gaussian))

4. One key assumption in your model is: “infection implies a reproductive cost for the host: everything else being equal, infected individuals generate less offspring than susceptible individuals.” While the relationship seems plausible, it would be helpful to clarify the basis for this assumption. Citing relevant references or empirical evidence to support it would strengthen the validity of the model and its underlying assumptions. Additionally, it would be beneficial to provide explanations/justifications for all assumptions made in the model.

5. In the text, it was mentioned that “agents starts with age 0 and are non-sexually mature until they reach the age of 1 day/1 month/1 year (i.e., 10 ticks)”. Could you clarify the basis for selecting 10 ticks as the threshold for sexual maturity?

6. The upper threshold for the age of death of agents was set at 300. I am wondering how the model outcomes might change if this threshold were set to a higher value, such as 1000. Providing a justification or basis for selecting this specific threshold would be helpful.

7. How did you validate your model? I believe this is a crucial aspect of the modeling process, as it ensures we can trust the implications the model provides. It may have been mentioned in the text but could benefit from further emphasis.

8. I am not sure I fully understood what you were trying to express in Equation (2). Should it be -N_i⋅i⋅o instead of -(N_i+i)⋅o?

Minor comments:

1. Line 143: Refer to the section discussing the scenarios

2. Line 219: think Equation (1) can be simplified as: N_0=100+N_m Δ_m, where Δ_m denotes the % difference in the benefits and costs of mating

3. In Tables 2A–2C, you might consider removing the 'Experimental Scenario' row and using the text in that row as the heading for the first column instead.

4. Line 303: Better to use the word replication instead of replica.

Reviewer #2: Major comments:

1) Not including a recovery state in the simulation is problematic since animals can recover from many STDs. For example, some animals can develop resistance to infections over time, and recovery can significantly impact disease dynamics. Including recovery rates would provide a more accurate representation of real-world scenarios and enhance the model's applicability. This should be included in the model and additional simulations should be made.

2) The reproductive benefit and cost percentages are good additions to the model. However, assuming fix values for these parameters is too simplistic. Additional simulations should be made using assumptions that are more applicable for these parameters. For example:

a) Trade-offs in Drosophila: Enhanced reproductive efforts in Drosophila result in a shorter lifespan, as the energy expended on courtship, egg laying, and mating takes a toll.

b) Parental Investment in Birds: Research on birds indicates that greater parental investment in current offspring can diminish the parents' future reproductive success and survival rates.

Specific Comments:

Line 171: "we consider a susceptible-infected system, without recovery for infected."

Comment: The model does not account for recovery from infection, which is a significant factor in the dynamics of many sexually transmitted diseases. Including recovery rates could provide a more accurate representation of disease dynamics.

Line 175: "Infection implies a reproductive cost for the host: everything else being equal, infected individuals generate less offspring than susceptible individuals."

Comment: The model assumes that the benefits and costs of mating are fixed percentages. In reality, these values can vary widely depending on environmental conditions, individual health, and other factors. This simplification might limit the applicability of the findings to real-world scenarios and even for hypothetical scenarios outlined by the model.

Line 146: "The key features of the model are applicable to a wide range of taxa with sexual reproduction, including many invertebrate and vertebrate species."

Comment: While the model is designed to be applicable to a wide range of taxa, the specific parameters and behaviors chosen may not accurately reflect the biology of all species. For example, the assumption that all males are polygynous might not hold true for species with different mating systems.

Line 169: "For the same reason, we did not create a further class of agents for the pathogen."

Comment: The model's reliance on a hypothetical pathogen means that the results may not be directly applicable to any specific real-world disease. The authors only considered an SI system without recovery which is not applicable to many sexually transmitted diseases.

Line 278: "We conducted Symmetric Analyses simulations, henceforth 'SA', where the initial number of M and P females is the same, that is, 50 agents for each strategy."

Comment: The impact of varying initial conditions, such as different starting ratios of males and females, and monandrous to polyandrous females, could be explored in more detail to understand how initial population structure influences the results.

Line 503: "Our agent-based modelling approach to the study of the interrelationship between mating strategies and epidemiological dynamics informs that this interplay may be remarkably complex and may have important repercussions for both the spread of disease and the evolution of sexual behaviours and mating systems."

Comment: The implications of the findings for managing sexually transmitted diseases in wildlife or human populations are not fully discussed. Addressing the ethical and practical considerations of applying these insights in real-world contexts would be valuable. Additionally, implying that a simple agent-based model has important repercussions must be backed up with model data validation from real-world data. This would enhance the credibility and applicability of the findings.

**Do you want your identity to be public for this peer review?** For information about this choice, including consent withdrawal, please see our Privacy Policy

Reviewer #1: No

Reviewer #2: No

---

## [Author Response · Author response to Decision Letter 1]

20 Feb 2025

Dear Editor,

You can find our complete letter and a point-by-point response to your comments and those of the reviewers in the document we named "Response to Reviewers".

Yours sincerely,

Riccardo Tarantino (corresponding author), on behalf of all authors

---

## [Decision Letter · Decision Letter 1]

7 Aug 2025

Dear Dr. Tarantino,

Thank you for submitting your manuscript to PLOS ONE. After careful consideration, we feel that it has merit but does not fully meet PLOS ONE’s publication criteria as it currently stands. Therefore, we invite you to submit a revised version of the manuscript that addresses the points raised during the review process.

We look forward to receiving your revised manuscript.

Kind regards,

Claus Kadelka

Academic Editor

PLOS ONE

Journal Requirements:

**Additional Editor Comments:**

I apologize for the slow review process. As you may be aware, we struggled securing reviews from the original referees. Reviewer #1 agreed but Reviewer #2 did not, requiring us to bring on a third reviewer. I agree with reviewer #3 that , in principle, your paper could be accepted with some revisions/corrections and clarifications. Your simulation work is sound but you draw conclusions that are too strong. There are also some issues with how the findings are communicated.

Reviewers' comments:

Reviewer's Responses to Questions

**Comments to the Author**

Reviewer #1: All comments have been addressed

Reviewer #3: (No Response)

2. Is the manuscript technically sound, and do the data support the conclusions?

Reviewer #1: Yes

Reviewer #3: Partly

3. Has the statistical analysis been performed appropriately and rigorously?

Reviewer #1: N/A

Reviewer #3: Yes

4. Have the authors made all data underlying the findings in their manuscript fully available?

Reviewer #1: Yes

Reviewer #3: Yes

5. Is the manuscript presented in an intelligible fashion and written in standard English?

Reviewer #1: Yes

Reviewer #3: Yes

Reviewer #1: All comments were addressed sufficiently. I do not have further comments on the manuscript. I think the current version of the manuscript can now be considered for publication in PLOS One.

Reviewer #3: This manuscript details a simulation of the relationship between female mating strategies and female-sterilising STDs. The manuscript then explores the how varying some aspects of the models parametrisation effects the success of a multi-mate versus single-mate strategy, and how this impacts the persistence of STDs and compares the two. These findings are interesting and publishable but there are issues with how these are communicated, I have divided my comments between major and minor issues that undermine the rigour of this analysis, and other comments that relate to the content but shouldn’t undermine this manuscript being published in PLOS One.

*Major Comments*

- A (acknowledged) limitation of this model is the lack of immunity or a “recovered” compartment in the SI and SIS structure. This is a fair assumption for many STIs but not everyone, for some reason the authors included fig 1 which attempts to list which STDs for which an SI or SIS model would be useful. It includes very questionable choices, such as HPV for SI and SIS and campylobacter, a confusing mix of vector-borne diseases that are STDs in vectors (and also in hosts) and is cited in a way that makes it very difficult to follow-up citations. Why include this table? It is completely unnecessary, the authors could have just provided examples to justify the use of SI and SIS model structures.

- The authors discuss STDs generally in the introduction and the title but the models exclusively focuses on STDs that might sterilise female infected, the title and introduction should be updated to reflect this. The discussion however does make this clear. In table 1 and the methods section so attempt is made to justify this choice (which is fair, many STDs are sterilising) however the citations chosen often refer to diseases which are diseases which also have detrimental impacts of the livelihood of infected animal and it raises why these aspects are ignored in the model. The authors should clarify their choices to focus on STDs that only impact fertility and correct their citations. Also some citations refer to the evolutionary cost of immune-system which is different to the costs of an infection, and not relevant to what the authors model captures.

- There is a comical trend amongst computational evolutionary biologists working on infectious diseases to rediscover R0 every time they publish a paper. R0 is well defined in endemic diseases (though often interpreted through Rt or Reff in relation to immunity). Just because your model is much simpler, this doesn’t make this the discovery of a new threshold. If you want to explore how this threshold relates to sexual strategy you could actually explore that by recording the number of sexual contacts in your model, it would be quite interesting.

- I don’t think that the conclusions on 825-829 are supported by your simulations. All the lower chance of success with a lower initial population indicates is that the smaller populations are more susceptible to stochastic effects. I’m not sure how the following line proceeds from that, please expand on your reasoning.

*Minor Comments*

- Some potential sensitivities are ignore and should be discussed or explored, in particular the impact of population threshold on model findings

- Simulation results are poorly plotted. The authors should consider including binomial proportion confidence intervals in their plots to account for the small simulation size used. The bar plots don’t need to depict both counts summing to 100, but should just show the proportion where M (or P) is dominant along with the confidence intervals. In general I find some of the comparisons unconvincing due to the small number of simulations so it would help support comparisons to have some measure of certainty given the simulation counts (or you could run more simulations). In addition the time to fixation plots within the same figure should be plotted on the same axis, i.e. fig2 B and D should both be plotted on a scale of 0-550 days/months/years. It would also be good to get an idea of the distributions around time to fixation using violin or boxplots rather than bar plots of the mean.

- I don’t understand the point of the equation f_m = 1/N on line 470?

- In figure 3 you compare the correlation of infections against the population of P females. Wouldn’t be to better to just compare the proportion of P females that were infected against the proportion of M females that were infected. I don’t understand why you’re using an abstraction here? You might need to adjust for the proportion of time at risk if there are STD/strategy extinctions but then you could just calculate the STD prevalence by female time(ticks) at risk. The same holds for Figure 4, why not plot the number of p-females infected and the number of m-females infected instead of comparing trends. You have a complete simulation you don’t need to use the limited approaches we have for observational data.

- How many simulations where used for Figure 6 and can we have some indication of the distribution of this?

- Figure 7, would be good to show the distributions of this from many simulations (plus a single line so we can have some idea of the trends within a single simulation). As an aside such a low level of prevalence for a life long disease suggests the number of mating events occur relatively slowly compared to the replenishment of susceptible, it would be interesting to see the sensitivity against the number of new born, I’d guess the results will change quite significantly.

*Other Comments (Not necessary for publication but for interest)*

- Would it not have been interesting to explore STDs that have a sterilising effect on males, i.e. wolbachia?

- I don’t really understand the value add of using a spatial structure here. This is mentioned in the introduction as the value of this paper over network based models but it seems to include no features that couldn’t be simulated with a network model (albeit with difficulty). A network model would allow you to include more realistic distributions of mating (i.e. heavy tails). This could do with some exploration in the model, i.e. what is the actual distribution of mating events etc in your model.

- This paper uses a lot of citations that don’t always directly address the claim being cited. These citations are often a decade or two old and are hard to access (even from a university with subscriptions to most journals). A single citation from a recent review or article (ideally from a open-access journal) would suffice and better represent the current state of the field.

- Why isn’t the cost parameter and the benefit parameter a single parameter? Can’t this just be impact of mating on offspring count, it seems needlessly confusing?

- Wouldn’t it be easier to express the time-frame in terms of generations or lifespan rather than constantly writing days/months/years?

- I found it quite hard to follow the interpretation of your results. It might be easier to read if you introduced the results and then discussed them before moving on the next set of simulations? Just a suggestion if that’s not against PLOS formatting rules.

Apologies if my comments seem harsh. In general the major issues are requests for you to clarify a couple of key-points or remove a couple of offending parts. In general I think this would help the make the paper more concise.

**Do you want your identity to be public for this peer review?** For information about this choice, including consent withdrawal, please see our Privacy Policy

Reviewer #1: **Yes: ** Allen Lamarca Nazareno

Reviewer #3: No

---

## [Author Response · Author response to Decision Letter 2]

22 Sep 2025

Dear Editor,

We are grateful for the opportunity to submit a new revision of our manuscript entitled "Agent-based modelling reveals feedback loops and non-linearity between mating system evolution and disease dynamics".

You can find our complete letter, together with a point-by-point response to Reviewer #3, in the document entitled "Response to Reviewers".

Best regards,

Riccardo Tarantino (corresponding author), on behalf of all authors

---

## [Decision Letter · Decision Letter 2]

21 Oct 2025

Agent-based modelling reveals feedback loops and non-linearity between mating system evolution and disease dynamics

PONE-D-24-14535R2

Dear Dr. Tarantino,

We’re pleased to inform you that your manuscript has been judged scientifically suitable for publication and will be formally accepted for publication once it meets all outstanding technical requirements.

Kind regards,

Claus Kadelka

Academic Editor

PLOS ONE

Additional Editor Comments (optional):

Reviewers' comments:

Reviewer's Responses to Questions

**Comments to the Author**

Reviewer #3: All comments have been addressed

2. Is the manuscript technically sound, and do the data support the conclusions?

Reviewer #3: Yes

3. Has the statistical analysis been performed appropriately and rigorously?

Reviewer #3: Yes

4. Have the authors made all data underlying the findings in their manuscript fully available?

Reviewer #3: Yes

5. Is the manuscript presented in an intelligible fashion and written in standard English?

Reviewer #3: Yes

Reviewer #3: I have reviewed the changes made to the manuscript, and the direct response to my comments. Overall, I am convinced by your even-handed responses and am happy with the changes made.

I am still unconvinced by the purpose of Figure 1, but this should not prevent publication any further.

Additionally, in regard to generation time I meant to use a discrete timescale that reflects the lifespan of the agents, i.e. one generation is 150 days, I should have been clearer, and this has no major impact on the paper.

For me personally, I found the combined result then discussion format much easier to follow and the additional analysis around the number of contacts interesting.

Thank you for the insightful research, and apologies for any inconvenience my delay in producing this review may have caused.

**Do you want your identity to be public for this peer review?** For information about this choice, including consent withdrawal, please see our Privacy Policy

Reviewer #3: No

---

## [Editor Report · Acceptance letter]

PONE-D-24-14535R2

PLOS ONE

Dear Dr. Tarantino,

I'm pleased to inform you that your manuscript has been deemed suitable for publication in PLOS ONE. Congratulations! Your manuscript is now being handed over to our production team.

Kind regards,

on behalf of

Dr. Claus Kadelka

Academic Editor

PLOS ONE